# Systematic design of superaerophobic nanotube-array electrode comprised of transition-metal sulfides for overall water splitting

Haoyi Li [1], Shuangming Chen[2], Ying Zhang[3], Qinghua Zhang[4], Xiaofan Jia[5], Qi Zhang[1], Lin Gu[4], Xiaoming Sun[3], Li Song[2] & Xun Wang[1]

Great attention has been focused on the design of electrocatalysts to enable electrochemical water splitting—a technology that allows energy derived from renewable resources to be stored in readily accessible and non-polluting chemical fuels. Herein we report a bifunctional nanotube-array electrode for water splitting in alkaline electrolyte. The electrode requires the overpotentials of 58 mV and 184 mV for hydrogen and oxygen evolution reactions respectively, meanwhile maintaining remarkable long-term durability. The prominent performance is due to the systematic optimization of chemical composition and geometric structure principally—that is, abundant electrocatalytic active sites, excellent conductivity of metallic 1T' $MoS_2$, synergistic effects among iron, cobalt, nickel ions, and the superaerophobicity of electrode surface for fast mass transfer. The electrode is also demonstrated to function as anode and cathode, simultaneously, delivering 10 mA cm$^{-2}$ at a cell voltage of 1.429 V. Our results demonstrate substantial improvement in the design of high-efficiency electrodes for water electrolysis.

[1] Key Lab of Organic Optoelectronics and Molecular Engineering, Department of Chemistry, Tsinghua University, Beijing 100084, China. [2] National Synchrotron Radiation Laboratory, CAS Center for Excellence in Nanoscience, University of Science and Technology of China, Hefei 230029, China. [3] State Key Laboratory of Chemical Resource Engineering, Beijing Advanced Innovation Center for soft Matter Science and Engineering, College of Energy, Beijing University of Chemical Technology, Beijing 100029, China. [4] Institute of Physics Chinese Academy of Sciences, Beijing National Laboratory for Condensed Matter Physics, Beijing 100190, China. [5] Department of Chemistry, University of Virginia, Charlottesville, VA 22904, USA. Correspondence and requests for materials should be addressed to X.W. (email: wangxun@mail.tsinghua.edu.cn)

Driven by the exigent demand for sustainable and renewable energy sources, a huge effort has been contributed to the development of efficient and accessible energy conversion technologies[1–3]. Electrochemical water splitting—a cost-effective and environmentally friendly technology for hydrogen evolution from electricity—has attracted great attention as a promising pathway to decrease social dependence on fossil fuels[4–6]. The synthesis and application of low-cost, high-performance electrocatalysts for water electrolysis are quite necessary to make water splitting become a functional and practical technology[7–9]. As a means of lowering the driving overpotential and increasing the efficiency of catalytic water splitting, there are several main aspects that deserve attention in the development of electrode materials. First, enhancing the intrinsic activity of catalytic sites in electrocatalysts requires an optimization of the Gibbs free energy of adsorption for reactants to speed the rate-determining step in the overall reactions[10–14]. Second, the enhanced conductivity of electrode materials is significant to enable fast charge transfer, a requirement for which metallic materials have demonstrated superior ability[15–17]. Third, the optimization of mass transfer properties has a tremendous effect on the ultimate efficiency of electrocatalysts. Innovative nanoarray architectures have great potential to facilitate the dissipation of as-generated gas bubbles from surface of the electrode and accelerate the reactions on cathode and anode—the hydrogen and oxygen evolution reaction (HER and OER)[18,19]. Finally, the increased density of electrocatalytic active sites in electrocatalysts allows for the maximization of limited electrode surface area—porous nanostructures have a proven ability to facilitate increased surface area[20,21].

With due consideration for the design principles enumerated above, we prepared a hybrid polymetallic sulfide nanotube-array electrode for water splitting via integrated compositional and geometric structural design. Metallic monoclinic 1T′ phase $MoS_2$ was employed for its excellent electrode kinetics, fast charge transfer, and remarkable intrinsic electrocatalytic HER activity[22–24]. Trimetallic iron, cobalt, nickel-based (Fe, Co, Ni-based) sulfides were utilized for their inherent and adjustable OER reactivity (a result of synergistic effects between metal ions)[25–27]. Finally, nanotube-array architectures were exploited for their porosity, high surface area, and uneven surface characteristics—a way to maximize catalytic active site density—and these morphological features provide the intriguing possibility of displacing as-generated gas bubbles from surface of the electrode simply—a feature which is known as superaerophobicity. Super-aerophobic behavior would allow greater contact between electrode and electrolyte, and thus promote efficient mass transfer[18].

In this work, the hybrid nanotube arrays (denoted as FeCoNi-HNTAs) are synthesized from ternary Fe, Co, Ni-based layered double hydroxide nanowire arrays (FeCoNi-LDH-NWAs) supported on Ni foam substrate. The FeCoNi-HNTAs show remarkable activity and long-playing durability in electrocatalytic

HER and OER. Notably, FeCoNi-HNTAs deliver a current density of 10 mA cm$^{-2}$ at an overpotential of 58 mV for the HER and 184 mV for the OER, respectively, while demonstrating outstanding durability (200 mA cm$^{-2}$ for 80-h continuous operation). The low Tafel slopes of 37.5 mV dec$^{-1}$ and 49.9 mV dec$^{-1}$ for HER and OER demonstrate the fast reaction kinetics. Additionally, we utilize FeCoNi-HNTAs in a water splitting system as anode and cathode simultaneously, achieving a current density of 10 mA cm$^{-2}$ at a cell voltage of 1.429 V in alkaline media. Synergistic effects among Fe, Co, and Ni ions are investigated by synchrotron radiation-based soft X-ray absorption spectroscopy (sXAS). Under-electrolyte superaerophobic and superhydrophilic features of the electrode surface are confirmed by adhesive force and contact angle measurements. Furthermore, ex situ and in situ synchrotron radiation-based extended X-ray absorption fine structure (EXAFS) characterizations are carried out to testify the presence and phase stability of 1T′ $MoS_2$.

## Results

**Synthesis and characterization of FeCoNi-HNTAs.** Taking the advantages into consideration, such as high surface area, superb electrical conductivity and low cost[17], Ni foam (Supplementary Fig. 1) was employed as the substrate for the synthesis of the expected electrode. FeCoNi-HNTAs, as schematically shown in Fig. 1, were synthesized through an accessible two-step solvothermal method (see details in Methods). First, FeCoNi-LDH-NWAs were prepared via a hydrolysis route[28,29]. Field-emission scanning electron microscope (FESEM) images (Fig. 2a) show the nanowire morphology bound to the substrate along with different directions and uniform diameter size of approximately 150 nm. The X-ray diffraction (XRD) pattern confirms the crystal structures of the as-prepared LDH nanowires (Supplementary Fig. 2). We also performed the energy-dispersive X-ray (EDX) elemental mapping measurement and found the uniform distribution of the five elements (Supplementary Fig. 3), which further indicated the successful synthesis of trimetal LDH nanowires. In the second step, FeCoNi-LDH-NWAs, regarded as the self-sacrificing templates, and ammonium tetrathiomolybdate (($NH_4$)$_2MoS_4$) were used as precursors for the synthetic reaction, where defined quantity of hydrazine hydrate (HZH) was also added. The FeCoNi-HNTAs were accessibly obtained in a large-scale level. Propagating the randomness of FeCoNi-LDH-NWAs, FeCoNi-HNTAs irregularly align on Ni foam and exhibit excellent dispersity and uniformity as shown in a larger scale image (Fig. 2b). The disorder contributes to higher porosity, thus exposing more catalytic active sites and facilitating mass transfer[30,31]. The structural generation process of the hybrid nanotubes from nanowires was studied (Supplementary Fig. 4) and the growth mechanism is considered as the Kirkendall cavitation[32,33]. The above synthetic strategy demonstrates a great achievement on morphology design for hybrid nanotube arrays. We further applied this method to the other two analogous systems

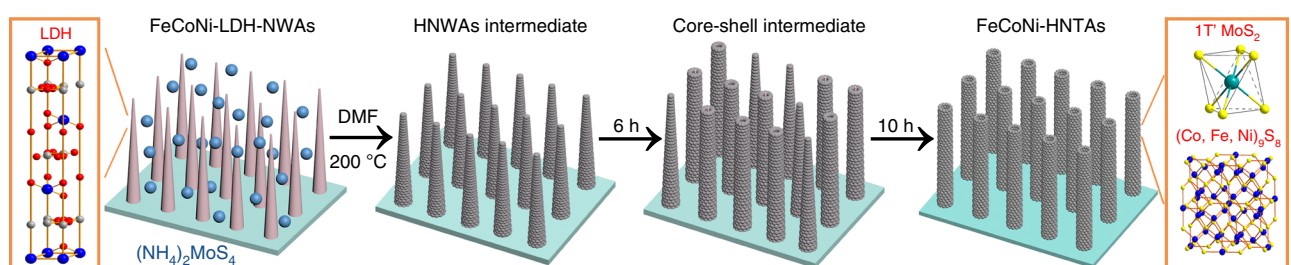

**Fig. 1** Schematic representation of the synthetic process of FeCoNi-HNTAs. FeCoNi-LDH-NWAs on Ni foam and ($NH_4$)$_2MoS_4$ are employed as the precursors with defined amount of HZH in the solvothermal synthesis of FeCoNi-HNTAs. FeCoNi-HNTAs form with crystalline structures of 1T′ $MoS_2$ and (Co,Fe,Ni)$_9S_8$ via the Kirkendall cavitation

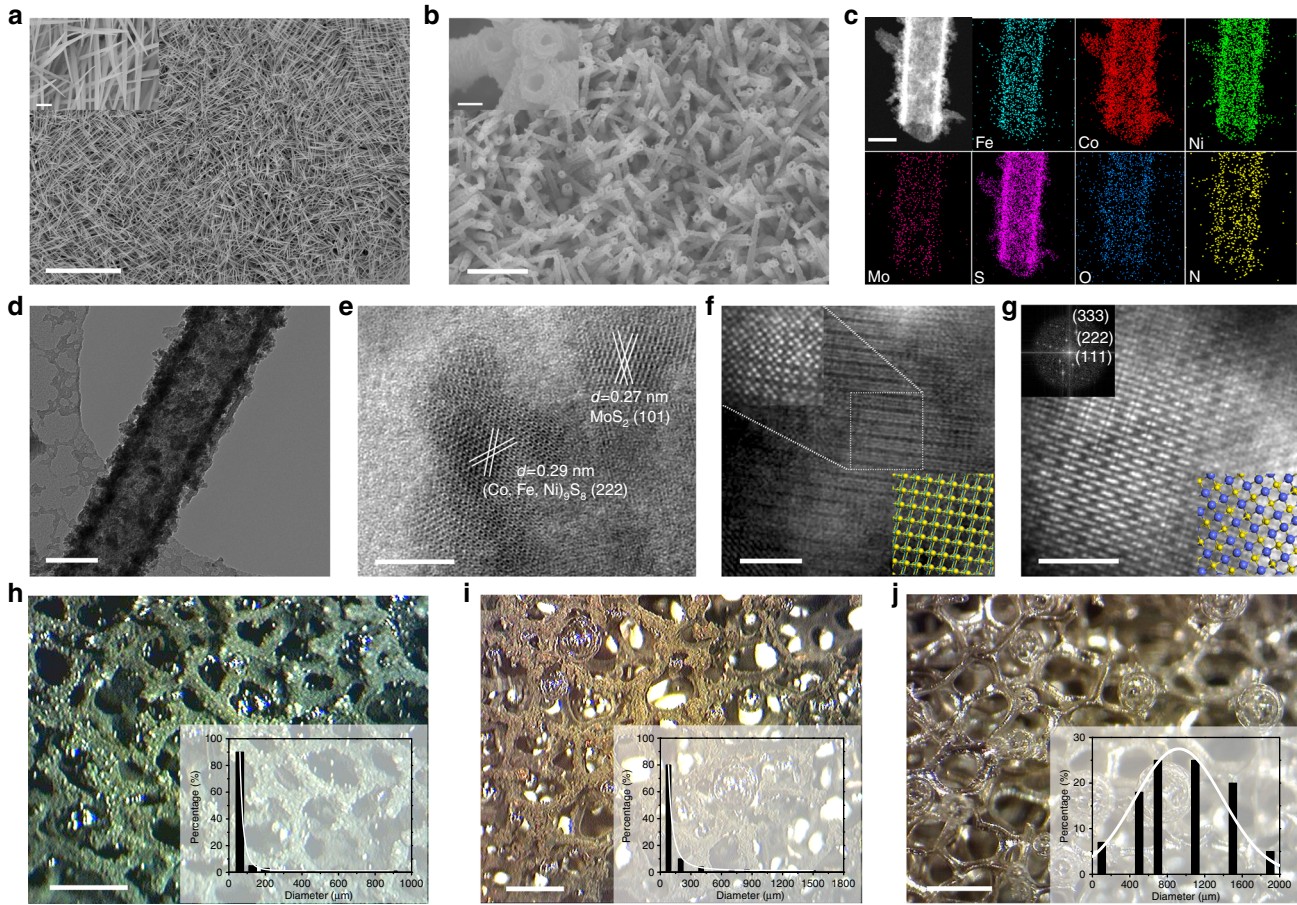

**Fig. 2** Morphological and structural characterizations and bubble releasing behaviors. **a**, **b** FESEM images demonstrating the uniformity of the as-prepared FeCoNi-LDH-NWAs and FeCoNi-HNTAs respectively as a large scale view. The insets of (**a**, **b**) are the corresponding magnified FESEM images and scale bars are 500 and 200 nm, respectively. **c** STEM and EDX elemental mapping spectra exhibiting the hollow nature, uneven tube wall, and uniform distribution of Fe (cyan), Co (red), Ni (green), Mo (pink), S (magenta), O (blue), and N (yellow) elements. **d** TEM image, **e** HRTEM image. The lattice fringes with the spacing of 0.27 nm and 0.29 nm are in agreement with the (101) and (222) planes of $MoS_2$ and $(Co,Fe,Ni)_9S_8$, respectively. **f**, **g** AC-STEM images of 1T' $MoS_2$ and $(Co,Fe,Ni)_9S_8$ in FeCoNi-HNTAs, respectively. The upper image of inset of **f** displays the Mo clusters with tetragonal symmetry marked by short dash line square in **f**. The corresponding FFT pattern of $(Co,Fe,Ni)_9S_8$ is shown as the upper image of inset of **g**. The lower images of the insets of **f**, **g** are the corresponding crystal structures, where yellow, cyan, and blue balls represent S, Mo, and Co (or Fe, Ni) atoms. **h**, **i**, **j** Digital photos demonstrating the bubble releasing behaviors on the surface of FeCoNi-HNTAs, FeCoNi-LDH-NWAs, and bare Ni foam for HER. The insets are the corresponding size distribution statistics of releasing bubbles. Scale bars: **a** 10 μm; **b** 2 μm; **c** 100 nm; **d** 100 nm; **e** 5 nm; **f**, **g** 2 nm; **h**, **i**, **j** 2 mm

(Supplementary Fig. 5). The hybrid nanotube and nanosheet arrays (CoNi-HNTAs and NiFe-HNSAs) synthesized from corresponding bimetal LDH nanoarrays, Co, Ni-based layered double hydroxides nanowire arrays (CoNi-LDH-NWAs) and Fe, Ni-based layered double hydroxides nanosheet arrays (NiFe-LDH-NSAs), verify the universality of this method. Furthermore, there were also two electrodes that were manufactured in the same method as FeCoNi-HNTAs for comparison (synthetic details shown in Methods). One is the electrode that $MoS_2$ directly grows on the Ni foam without the precursor of FeCoNi-LDH-NWAs (Supplementary Fig. 6a), denoted as $MoS_2$/Ni Foam. The other is the electrode showing porous nanotube array morphology that FeCoNi-LDH-NWAs are sulfurated by thioacetamide (TAA) instead of $(NH_4)_2MoS_4$ (Supplementary Fig. 6b), denoted as FeCoNiS-NTAs.

When focusing on a single nanotube in FeCoNi-HNTAs (Fig. 2b, c, d), the diameter size is around 150 nm and the thickness of the tube wall ranges from approximately 30 to 80 nm. Hollow characteristic and high degree of surface roughness can be distinctly observed (Supplementary Fig. 7), which proliferate the active surface area and expose more catalytic active sites.

Meanwhile, EDX elemental mapping spectra (Fig. 2c) elucidate the elementary composition of FeCoNi-HNTAs. Besides, the presented seven elements uniformly distribute in a single tube. We also carried out the EDX spectra (Supplementary Fig. 8) to illustrate the elemental ratios, that showed Fe: Co: Ni was about 1:5:3 and the percent proportions of Mo in FeCoNi-HNTAs reached about 11.9%, which was consistent with the feed ratios. Moreover, integrating high-resolution transmission electron microscopy (HRTEM) image (Fig. 2e) and corresponding XRD pattern (Supplementary Fig. 9), the characteristic lattice fringes of (101) planes in $MoS_2$ with inter-planar distances of 0.27 nm were exhibited as well as (222) planes of $(Co,Fe,Ni)_9S_8$ with inter-planar spacing of 0.29 nm, which is confirmed by the crystalline nature of FeCoNi-HNTAs. The HRTEM image further demonstrates successful formation of the hybrid nanostructures that presents the combination and no boundaries of the two compositions. Typical atomic resolution aberration-corrected scanning TEM (AC-STEM) images clarify the presence of 1T' $MoS_2$ and $(Co,Fe,Ni)_9S_8$, respectively. The superlattices of 1T' $MoS_2$ with Mo clusters could be clearly visualized in Fig. 2f, distinctly exhibiting expected tetragonal symmetry of atomic

arrangement[34]. Cubic symmetry of the crystals can also be apparently distinguished in Fig. 2g and the corresponding fast Fourier transform (FFT) pattern is indexed unequivocally to (Co, Fe,Ni)$_9$S$_8$[35]. These characterization results demonstrate our compositional and structural design for the hybrid system, whose superiorities mentioned above are considered to be the stepping-stone for the enhancement of electrocatalytic performance.

In addition to the construction of chemical compositions, visual behavior of as-generated gas bubbles releasing from the surface of FeCoNi-HNTAs demonstrates the morphological advancements of FeCoNi-HNTAs in electrocatalytic water splitting. Specifically, the bubble releasing behaviors during HER and OER can directly reflect the mass transfer efficiency[18]. A high-speed camera with a charge-coupled device recorded the bubble release during a galvanostatic scan for HER and OER at the current density of 20 mA cm$^{-2}$. As shown in Fig. 2h, i, j, distinct comparison between FeCoNi-HNTAs, FeCoNi-LDH-NWAs and bare Ni foam for as-formed gas bubbles releasing behaviors proves superior mass transfer on nanoarray electrode than the flat electrode, where the releasing bubbles have the major size distribution of less than 100 μm and around 1000 μm, respectively. Other bubble releasing phenomena are displayed in Supplementary Fig. 10. The quick releasing process of gas bubbles on nanoarray electrode verifies that the architectures we developed possesses quite fascinating surface feature for fast mass transfer, suggesting that it is a promising candidate for scale-up utilization in practical applications.

In-depth characterizations were carried out to illustrate the fine structures and chemical valence states of the hybrid nanostructures. In this solvothermal system, we used HZH as inductive agent and electron donor to achieve the formation of metallic 1T' MoS$_2$[21]. To identify its phase, the local bond nature was investigated by EXAFS at Mo K-edge. According to the normalized X-ray absorption near edge structure (XANES) spectrum from EXAFS (Fig. 3a), the pre-edge peak of FeCoNi-HNTAs shifts to higher energy than that of the standard 2H MoS$_2$ foil, that is due to larger oxidation degree of MoS$_2$. In the meantime, MoS$_2$ in FeCoNi-HNTAs possesses obvious difference of white line peak, suggesting the different local atomic arrangements. The FT profile comparison of EXAFS data at Mo K-edge in R-space (Fig. 3b) exhibits distinct downshift of Mo-Mo bond peak in FeCoNi-HNTAs, demonstrating decreasing bond length from 3.16 to 2.79 Å (Supplementary Table 1), which corresponds to the characteristic peak of Mo–Mo bond in 1T' MoS$_2$[36]. This result also proves the stability of 1T' MoS$_2$ in FeCoNi-HNTAs under ambient environment (Supplementary Note 1). Besides, the evidently higher intensity of Mo–O/N bond peak should be due to the coordination with HZH and slight oxidation from air. The fitting curve for FT profiles is exhibited in Supplementary Fig. 11, which matches well with the experimental data. The X-ray photoelectron spectroscopy (XPS) spectrum (Fig. 3c) of Mo 3d and S 2 s regions of MoS$_2$ in FeCoNi-HNTAs shows the fitting curve from characteristic peaks of 1T' (228 and 231.2 eV) and 2H phase (229 and 232 eV)[37], resulting in the major proportion of 1T' MoS$_2$ in FeCoNi-HNTAs. The above illustration completely testifies the MoS$_2$ in FeCoNi-HNTAs as 1T' phase, which can provide higher density of electrocatalytic active sites for HER and greatly facilitate electrical conductivity of the electrode. Moreover, the high-resolution XPS spectra comparison (Supplementary Fig. 12) was performed to demonstrate the interaction between the two compositions in FeCoNi-HNTAs. Compared with that of MoS$_2$/Ni Foam, the downshift about 0.8 eV of the peak positions of FeCoNi-HNTAs can be obviously observed, which elucidates the charge transfer from Fe, Co, Ni-based sulfides to MoS$_2$ to facilitate the phase transition from 2H to 1T' phase[21,37]. This interaction between the two compositions further indicates the hybrid material nature of FeCoNi-HNTAs.

The chemical valence states of Fe, Co, and Ni in FeCoNi-HNTAs was investigated by high-resolution XPS[38]. As shown in Fig. 3d, there are three kinds of characteristic peaks, belonging to Fe-S (706.7 eV)[39], Fe$^{2+}$ (710.4 eV), and Fe$^{3+}$ (712.8 eV) 2p regions, that enable the fitting curve to match the original signal well. The broad peak representing Fe$^{2+}$ and Fe$^{3+}$ demonstrates the presence of oxidation states of iron. As for the XPS spectrum of Co 2p regions (Fig. 3e), the main peak can be split into two distinctive peaks of Co$^{3+}$ (780.7 eV) and Co$^{2+}$ (782.1 eV), of which the shoulder at lower binding energy is corresponding to the Co-S peak (778.0 eV)[40]. Similarly, both of Ni$^{2+}$ (855.0 eV) and Ni$^{3+}$ (856.1 eV) could be measured in the final product according to the XPS spectrum of Ni 2p regions (Fig. 3f). Therefore, it is revealed that the high and low oxidation states of Fe, Co, and Ni coexist in FeCoNi-HNTAs.

For the first row transition metal-based compounds, sXAS is performed for direct detection of the 3d valence states of the metals via dipole selection rules. The 2p core electrons enable the excitation to be transferred into the empty 3d orbitals. This transition process allows the exploration of the 3d electrons with high-intensity, which is beneficial to acquire a comprehensive understanding and systematically optimize materials for electrocatalysis. Here we mainly focus on the L$_3$ regions at Fe, Co, and Ni L-edge sXAS recorded in the total electron yield (TEY) mode to illustrate the interaction among them. At Fe L$_3$ edge (Fig. 3g), there are two groups of peaks around 706.7 eV and 708.1 eV, corresponding to the fingerprints of Fe$^{2+}$ and Fe$^{3+}$, respectively[41]. Compared with NiFe-HNSAs, the amount of Fe$^{3+}$ increases obviously in FeCoNi-HNTAs, indicating Co ions can adjust the electronic state of Fe to favor the high valence state. Friebel et al. have reported that Fe$^{3+}$ sites are indeed highly active for OER, because the intermediate of OER process has quite low Gibbs free energy of adsorption on Fe sites. This result indicates that Co behavior has a positive effect on the enhancement of OER activity[25]. When referring to Fe incorporation impact, the comparison of Co L-edge sXAS between FeCoNi-HNTAs and CoNi-HNTAs is exhibited in Fig. 3h. A small shoulder at 780.4 eV of FeCoNi-HNTAs is owing to the distinctive peak of the Co$^{3+}$ ions located at octahedral sites, which demonstrates that the participation of Fe in our hybrid system can tune the crystal-field coordination of Co ions. It have been investigated that Co$^{3+}$ ions located at octahedral sites are the active centers for OER rather than the Co$^{2+}$ ions at tetrahedral sites (778.2 eV) and the interaction between Fe and Co ions is considered to improve the intrinsic activity for OER and affects chemical and structural stability[26,42,43]. Therefore, the synergetic effect of Fe and Co ions in our hybrid system is beneficial for the enhancement of OER performance. Besides, Fe and Co are demonstrated to influence the local electronic structure of Ni cations according to the Ni L-edge sXAS (Fig. 3i). In contrast with NiFe-HNSAs and CoNi-HNTAs, FeCoNi-HNTAs apparently show a shoulder peak at 852.8 eV in agreement with the fingerprint of Ni$^{3+}$, which is regarded as electrocatalytic sites with higher activity than Ni$^{2+}$ (sXAS L$_3$-edge peak at 850.7 eV) for OER[25,44]. Thus the interaction between Fe, Co, and Ni in our hybrid system has a great significance on the improvement of OER electrocatalytic activity. Based on the analyses of the fine structures and chemical valence states of the FeCoNi-HNTAs, it is summarized that synergistic effects can be verified among Fe, Co, and Ni ions, which is extraordinarily critical to promote OER electrocatalysis.

**Under-electrolyte superaerophobicity of FeCoNi-HNTAs.** To identify the origin of nanoarray structural advantages for bubble release from the electrode surface with ease, surface super-aerophobicity of the as-prepared electrodes was investigated by

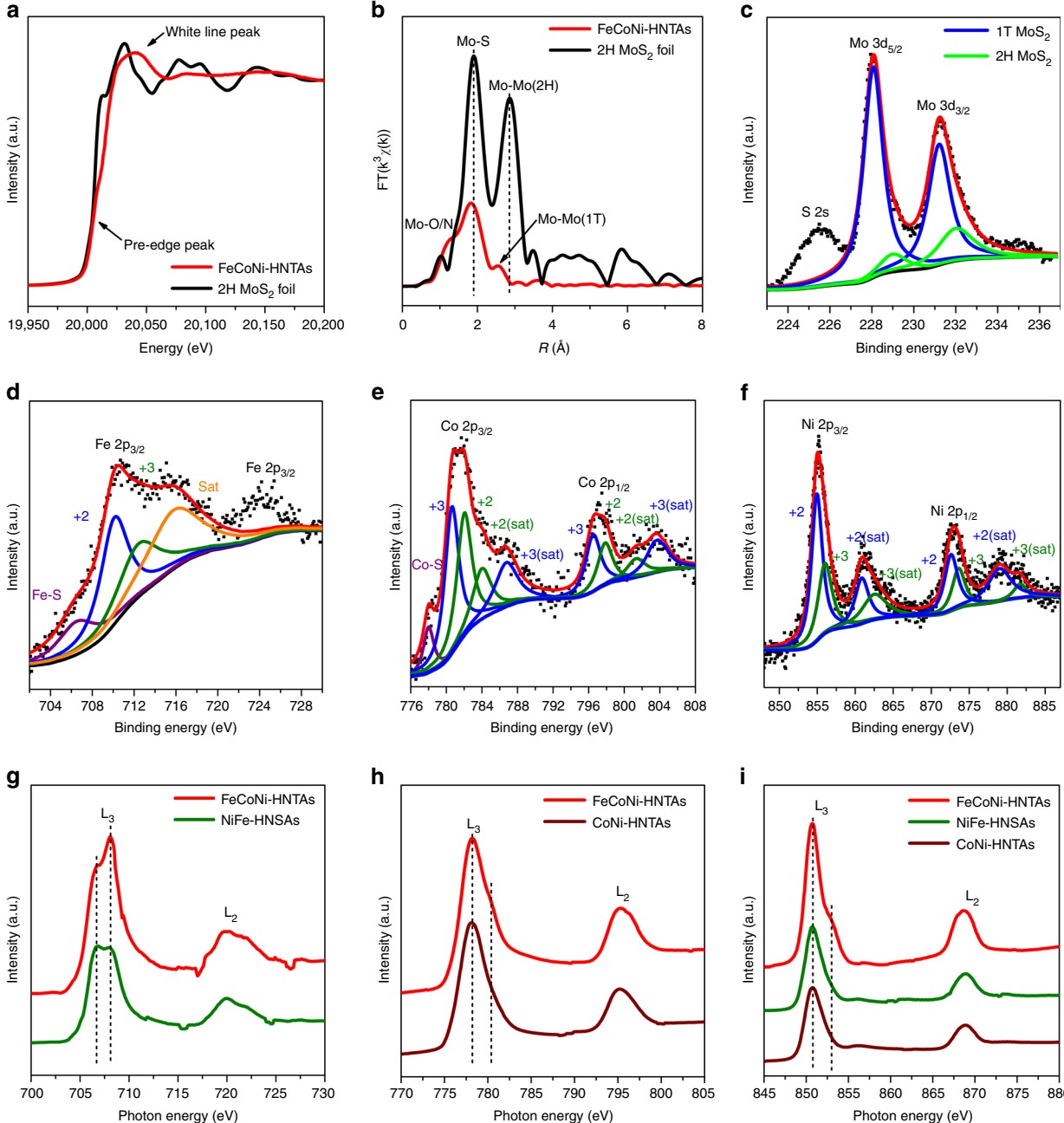

**Fig. 3** In-depth compositional and structural analyses of FeCoNi-HNTAs. **a**, **b** The normalized ex situ XANES spectra and corresponding $k^3$-weighted FT profile in R-space from EXAFS at Mo K-edge of FeCoNi-HNTAs. **c**–**f** High-resolution XPS spectra of Mo 3d, Fe 2p, Co 2p, and Ni 2p regions with fitting curves. **g**–**i** Normalized Fe, Co, and Ni L-edge sXAS spectra of FeCoNi-HNTAs and the counterparts in TEY mode

adhesive force of the bubbles to the surface and bubble contact angle measurements in 1 M KOH media (applied electrolyte for electrocatalytic measurements), in which the wetting ability or contact manner on the under-electrolyte surface is crucial[45]. As shown in Fig. 4a, there is not any adhesive force between the bubble and the surface measured for FeCoNi-HNTAs, which is further demonstrated through the inappreciable deformation of the bubble in the process of the measurement (inset 1–3 of Fig. 4a). Meanwhile, the bubble contact angle reaches 171.0° ± 3.4° on the surface of FeCoNi-HNTAs under electrolyte (inset 4 in Fig. 4a), indicating the remarkable superaerophobicity of FeCoNi-HNTAs. This feature can be attributed to the discontinuous state of the three-phase contact line (TPCL) of the bubbles with surface of the electrode, contributing to the

exceptionally low contact region between the bubbles and surface of the electrode and thus low adhesive force[18]. Surface superhydrophilicity of FeCoNi-HNTAs is also displayed in inset 5 of Fig. 4a. The electrolyte droplet infiltrated the electrode immediately and could not be captured by high-speed camera. This excellent superhydrophilicity results in the formation of a wetting film on the surface of the electrode. By increasing the amount of electrolyte, the electrolyte liquid film may reduce the contact region between the bubbles and surface of the electrode and hence decrease the bubble adhesive force. Employing FeCoNi-LDH-NWAs, MoS₂/Ni Foam, bare Ni foam, FeCoNi-HNWAs, CoNi-HNTAs, NiFe-HNSAs, and FeCoNiS-NTAs as contrastive samples (Fig. 4b, c, d, and Supplementary Fig. 13), the values of adhesive force, bubble contact angle under electrolyte, the

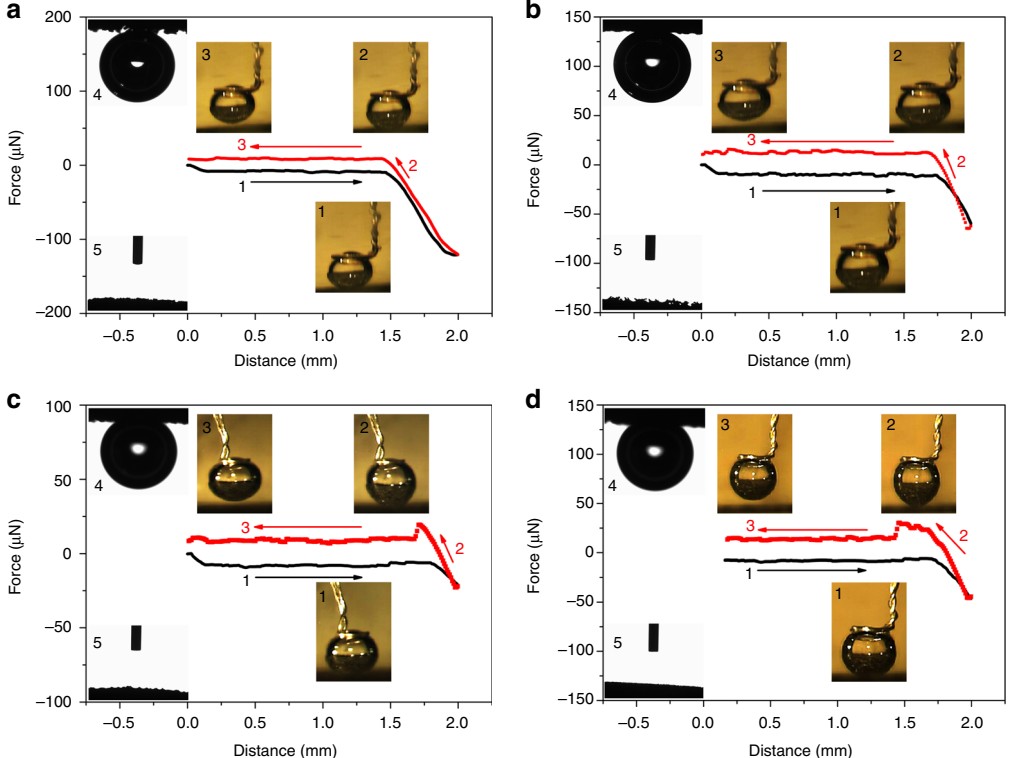

**Fig. 4** Under-electrolyte superaerophobic and superhydrophilic measurements. **a–d** Gas bubble adhesive force measurements of FeCoNi-HNTAs, FeCoNi-LDH-NWAs, MoS$_2$/Ni Foam, and bare Ni foam respectively. FeCoNi-HNTAs and FeCoNi-LDH-NWAs obviously show no bubble adhesive force while the larger bubble adhesive forces are measured on MoS$_2$/Ni Foam (10.6 ± 1.9 μN) and bare Ni foam (16.0 ± 1.7 μN). The insets 1–3 show the bubble states in the corresponding measurement process of adhesive force where process 1 illustrates the electrode surface gets close to the air bubble, process 2 demonstrates the electrode surface leaves the air bubble and process 3 displays the electrode surface separates from the air bubble. The distinct bubble deformation observed in insets 2 of **c** and **d** further prove the adhesivity of MoS$_2$/Ni Foam and bare Ni foam to bubbles. For insets 4, the bubble contact angles under electrolyte were measured as 171.0° ± 3.4°, 156.5° ± 2.9°, 146.2° ± 2.1°, and 143.7° ± 2.3° respectively. Insets 5 show the wetting ability of the electrodes in which KOH solution droplets can not be captured suggesting the surface superhydrophilicity of the employed electrodes

advancing angle, the receding angle, the corresponding contact angle hysteresis and electrolyte droplets contact angle are all summarized in Supplementary Table 2. It is observed that all of the nanoarray electrodes we explored show prominent super-aerophobicity and superhydrophilicity, which manifests the deployed nanoarray electrodes are indeed preponderant in electrocatalysis of gas evolution reaction. The decrease of roughness is embodied in MoS$_2$/Ni Foam and bare Ni foam (Supplementary Fig. 6a and Supplementary Fig. 1), which have bulk morphology with large size (more than 500 nm) and flat surface respectively. The Wenzel–Cassie transition demonstrates a continuous change of surface wetting behavior with the change of gas contact mode on the interface based on the roughness increase[45]. Therefore, MoS$_2$/Ni Foam and bare Ni foam possess enhanced adhesion ability to bubbles in the solution. We believe that the above consequences can inspire progress in specific architecture of nanoarray electrode with desirable surface superaerophobicity and wetting state, leading to efficient bubble release from the electrode surface for gas evolution reaction.

**Evaluation of electrocatalytic water splitting performance.** The as-prepared electrodes were directly applied to a typical three-electrode cell as the working electrode for HER and OER electrocatalysis in 1 M KOH electrolyte. The Ag/AgCl electrode was utilized as the reference electrode and a graphite rod as the counter electrode. Reversible hydrogen electrode (RHE) calibration was also performed (Supplementary Fig. 14) and all of the

potentials in the measurements were exhibited vs. RHE. The incipient measurement was performed via cyclic voltammetry for activation and stabilization of the electrodes. For HER, linear sweep voltammetry was first carried out for recording the polarization ability of the as-synthesized electrodes. FeCoNi-HNTAs shows remarkably enhanced catalytic activity compared to other employed electrodes (Fig. 5a and Supplementary Fig. 15a), which just needs the overpotentials ($\eta_{HER}$) of 58 mV to afford a current density of 10 mA cm$^{-2}$ ($\eta_{HER}$ for FeCoNi-LDH-NWAs, MoS$_2$/Ni Foam, FeCoNiS-NTAs, bare Ni foam, FeCoNi-HNWAs, CoNi-HNTAs, and NiFe-HNSAs are 214 mV, 181 mV, 121 mV, 322 mV, 80 mV, 70 mV, and 169 mV, respectively). Notably, the HER performance of FeCoNi-HNTAs approaches that of commercial Pt/C catalyst loaded on Ni foam, in which the difference of $\eta_{HER}$ is only 11 mV. The catalytic reaction kinetics is studied by Tafel plot. As it is shown in Fig. 5b, the slope value of Tafel curve of FeCoNi-HNTAs exhibits 37.5 mV dec$^{-1}$, smaller than that of FeCoNiS-NTAs (59.2 mV dec$^{-1}$), FeCoNi-LDH-NWAs (122.6 mV dec$^{-1}$), and MoS$_2$/Ni Foam (95.2 mV dec$^{-1}$), verifying the rapid HER catalytic rate. This Tafel slope value recorded on FeCoNi-HNTAs also indicates better inherent catalytic activity for HER from the perspective of reaction mechanism[24]. Supported by conductive substrate, all of the electrodes possess a low level of charge transfer resistance (Supplementary Fig. 16a) proven by the Nyquist plots of electrochemical impedance spectroscopy (EIS). Compared with other contrastive samples, the resistance of FeCoNi-HNTAs is smaller by an order of magnitude due to the composition of metallic 1T' MoS$_2$. To

assess the electrochemical active surface area (ECSA), double-layer capacitance ($C_{dl}$) was measured to roughly calculate the value of ECSA. The current densities at the selected potential from the regions of no Faradaic processes were showed as the linear correlation with the scan rates (Fig. 5c and Supplementary Fig. 17a, c, e, g) and the slopes of the fitting curves were considered as the $C_{dl}$. FeCoNi-HNTAs have the largest ECSA value of 3577.5 cm$^2$ among the four samples, suggesting the most

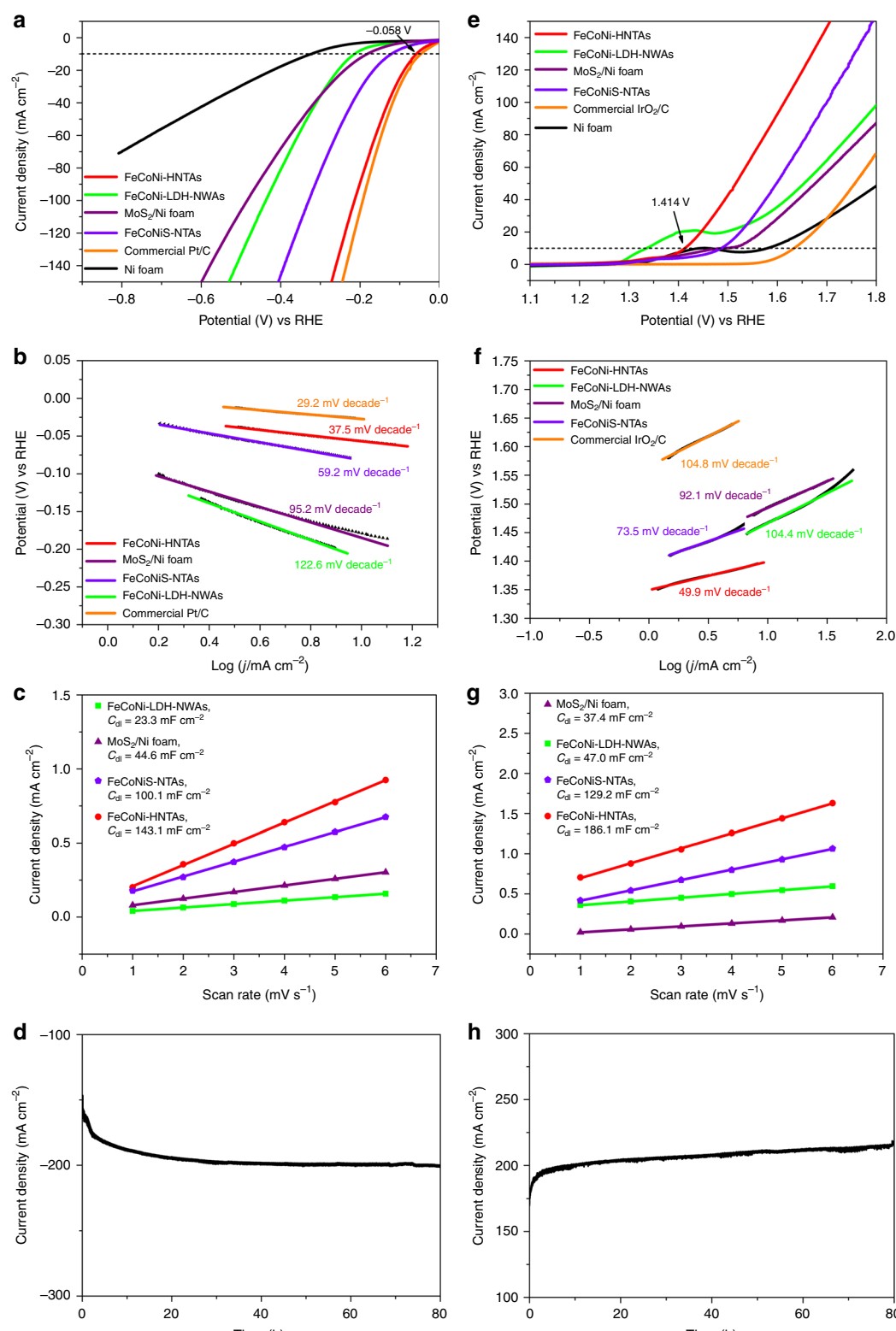

**Fig. 5** HER and OER electrocatalytic performance. **a**, **e** Polarization plots for HER and OER processes in 1 M KOH electrolyte. Scan rate: 1 mV s$^{-1}$. **b**, **f** Tafel curves for HER and OER processes showing the reaction kinetics. Scan rate: 1 mV s$^{-1}$. **c**, **g** $C_{dl}$ measurements for HER and OER processes. **d**, **h** Chronoamperometric measurements (*i–t*) recorded on FeCoNi-HNTAs for 80 h at the steady potentials of −0.326 V vs. RHE for HER and 1.796 V vs. RHE for OER, respectively

catalytic active sites for HER. As for the durability of FeCoNi-HNTAs, we carried out two sorts of measurements via chronoamperometry and galvanic pulse method to elucidate the continuous polarization ability and structural recoverability[27]. Figure 5d illustrates almost steady current density of near 200 mA cm$^{-2}$ during an 80-h chronoamperometry running. Meanwhile, periodic galvanic pulses recorded on FeCoNi-HNTAs at two different current density of 100 mA cm$^{-2}$ and 200 mA cm$^{-2}$ can be sustained for 30 h (Supplementary Fig. 18a). Apparently, the current density maintains almost unchanged, indicating the fascinating structural recoverability of FeCoNi-HNTAs due to the intrinsic structural superiority of hollow morphology. Moreover, Faradaic efficiency of the HER process over FeCoNi-HNTAs should also be evaluated as a crucial performance index. We employed a gas chromatography to monitor the amount of evolved H$_2$ and the theoretical quantity of H$_2$ at the current density of 20 mA cm$^{-2}$ for 120 min continuously was calculated by the Faraday law. The comparison between the theoretical and experimental data (Supplementary Fig. 19a and Supplementary Table 3) presents a satisfying Faradaic efficiency of 99.7% for HER process, demonstrating that the measured electrocatalytic cathodic current stems exclusively from water reduction.

As the major barrier of water splitting, OER hinders efficiency promotion of overall water splitting. A high-performance electrode plays a pivotal role in overcoming the large overpotential to drive the reaction. We also used FeCoNi-HNTAs to electrocatalyze OER and thus assess the performance. FeCoNi-HNTAs demonstrates the overpotential ($\eta_{OER}$) at the current density of 10 mA cm$^{-2}$ can reach as low as 184 mV (Fig. 5e), which achieves a great success as an electrocatalyst with low overpotential for OER. Markedly, polarization capability of FeCoNi-HNTAs is much higher than other contrastive samples (Supplementary Fig. 15b), especially for commercial IrO$_2$/C ($\eta_{OER} \approx 400$ mV). Faster catalytic reaction kinetics and electron transfer over FeCoNi-HNTAs are demonstrated by low Tafel slope (49.9 mV dec$^{-1}$, Fig. 5f) and reaction resistance (≈1 ohm, Supplementary Fig. 16b). In the meantime, the ECSA for OER is estimated by the value of $C_{dl}$ (Fig. 5g and Supplementary Fig. 17b, d, f, h). The higher ECSA of FeCoNi-HNTAs, 4652.5 cm$^2$, further proves the compositional and structural advantages in exposure of catalytic active sites. Moreover, FeCoNi-HNTAs could maintain 80-h activity for holding the current density of around 200 mA cm$^{-2}$ with slight increase (Fig. 5h) and 30-h structural recoverability during recurrent galvanic pulses between 100 and 200 mA cm$^{-2}$ (Supplementary Fig. 18b). To prove the percent conversion from anodic current to O$_2$ on FeCoNi-HNTAs, OER Faradaic efficiency was studied (Supplementary Fig. 19b and

Supplementary Table 3) and exhibited a reasonable value of 90.9%, which was attributed to the complicated process of four-electron transfer and the existence of oxidation current.

After 1000 HER and OER cycles, we collected and characterized the FeCoNi-HNTAs electrodes to identify the compositional and structural changes. As it is shown in Supplementary Fig. 20, it is unchanged for the crystal structures of FeCoNi-HNTAs after HER and OER cycles, similar to the as-prepared sample. SEM images (Supplementary Fig. 21a, b) show that nanotube array structures and uneven surfaces can be maintained and the size of a single nanotube stays the same as the initial one. The (101) planes of 1T' MoS$_2$ with aligned arrangement and the (222) lattice fringes of (Co,Fe,Ni)$_9$S$_8$ are observed in HRTEM images (Supplementary Fig. 21c, d), in accordance with the corresponding XRD patterns. EDX mapping spectra (Supplementary Fig. 22) further confirmed the uniform distribution of the seven elements. Compared with the as-prepared sample, the ratios of elements illustrated by EDX spectra (Supplementary Fig. 23) barely changed except for slightly increasing amount of oxygen after OER cycles. Moreover, the comparisons of smoothing XPS spectra of Fe, Co, and Ni 2p regions of FeCoNi-HNTAs before and after electrochemical measurements (Supplementary Fig. 24a, b, c) were made for the changes of Fe, Co, and Ni chemical states. As a result, HER process is conducive to the stabilization of valence states and there are hardly any shifts for Fe, Co, and Ni 2p peaks. However, small upshifts (about 0.3 eV) are observed for the peaks of Fe, Co, and Ni 2p regions of FeCoNi-HNTAs after OER cycles, demonstrating in situ low-level oxidation of Fe, Co, Ni-based sulfides in the OER process, which is accordance with the EDX results. As for 1T' MoS$_2$, the two distinctive peaks of Mo 3d$_{5/2}$ and 3d$_{3/2}$ regions remain the same positions after HER and OER cycles (Supplementary Fig. 24d), which directly proves that this hybrid system plays a vital role in stabilizing 1T' MoS$_2$[21]. In accordance with the above results of characterizations after electrochemical measurements, this hybrid system is quite robust and practical for electrocatalytic process of water oxidation and reduction, raising the hope of industrialization.

As a promising bifunctional electrocatalyst, FeCoNi-HNTAs were also employed to function as cathode and anode simultaneously incorporated with a configuration containing 1 M KOH media for overall water splitting (OWS) at room temperature. Impressively, the electrodes exhibit prominent activity in this two-electrode system, requiring 1.429 V ($E_{OWS}$) of the cell voltage to drive a current density of 10 mA cm$^{-2}$ (Fig. 6a), which demonstrates superior electrocatalytic activity than other electrodes (Supplementary Fig. 25). Compared with commercial IrO$_2$/C–Pt/C couple ($E_{OWS} \approx 1.58$ V),

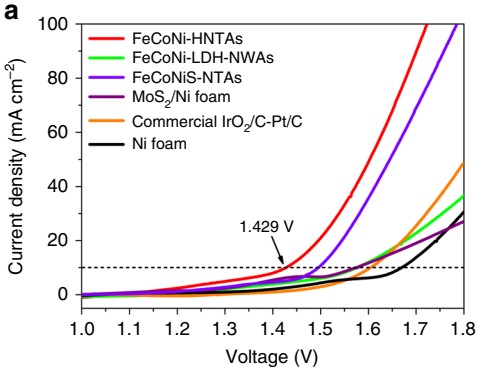
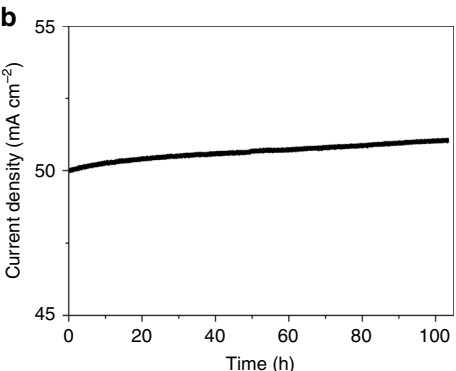

**Fig. 6** Evaluation of the performance over FeCoNi-HNTAs for overall water splitting. **a** Polarization plots in 1 M KOH electrolyte with a typical two-electrode set-up. Scan rate: 1 mV s$^{-1}$. **b** Chronoamperometric response recorded on FeCoNi-HNTAs at a constant cell voltage of 1.59 V

FeCoNi-HNTAs elucidate the overwhelming activity and excellent ability for high-current operations. The long-playing stability measurement (Fig. 6b), maintaining current density of about 50 mA cm$^{-2}$ for more than 100 h, illustrates the great potential of FeCoNi-HNTAs for commercial utilization to replace noble-metal materials, which show a continuous degradation for a constant cell voltage operation (Supplementary Fig. 26). The overall water splitting process via chronoamperometric method at 1.55 V of the cell voltage for about 1.5 min was recorded in Supplementary Movie 1, where H$_2$ and O$_2$ bubbles escaped continuously from surface of the left and the right electrode, respectively. This applicable testing offers a prospective evaluation of the feasibility for applying FeCoNi-HNTAs electrode into practical water splitting system.

**EXAFS analysis for phase stability of 1T' MoS$_2$.** It has been evidenced that metallic 1T' MoS$_2$ is metastable with higher ground-state energy than its semiconducting 2H counterpart[46]. If there are some destabilizing factors affecting the 1T' phase, such as high temperature, it preferentially transforms into 2H phase, which is thermodynamically stable[47]. Therefore, we constructed this hybrid nanostructure to stabilize 1T' MoS$_2$, making it provide persistent functionality during electrocatalytic processes. To clarify the stability of 1T' MoS$_2$ in HER and OER electrocatalysis, we performed both ex situ and in situ EXAFS at Mo K-edge for FeCoNi-HNTAs. The in situ XANES spectra from EXAFS (Fig. 7a, c) and corresponding FT profiles (Fig. 7b, d) in R-space of FeCoNi-HNTAs at as-prepared and ongoing electrocatalytic states were carried out at different potentials during potentiostatic HER and OER measurements. It is noteworthy that the positions

of Mo-Mo bond peaks for 1T' MoS$_2$ are unchanged during the applied HER and OER processes compared to that of the initial sample, indicating 1T' MoS$_2$ in our hybrid system enables splendid stability at both reduction and oxidation potentials. According to ex situ EXAFS experiments (Supplementary Fig. 27, 28, Supplementary Table 1), MoS$_2$ in FeCoNi-HNTAs after galvanostatic scan of HER and OER (Supplementary Note 1) still kept 1T' phase, which was in accordance with the in situ EXAFS data. However, OER operations bring about the increasing proportion of oxidized MoS$_2$ that are positively related to the potentials. At the meantime, long-term electrolysis running can also generate oxidized MoS$_2$, suggesting that 1T' MoS$_2$ affects both half reactions of water splitting. In this system, FeCoNi-LDH-NWAs can coordinate with HZH to form complexes for protection of 1T' MoS$_2$[21]. Supplementary Fig. 29 exhibits the representative core-shell nanostructure after 1000 OER cycles. Integrated with the corresponding XRD pattern (Supplementary Fig. 18), the core can be indexed to 1T' MoS$_2$ with the zigzag chains and the shell matching with the amorphous component layer. This strategy has been successfully developed for stabilizing 1T' MoS$_2$, which can be considered as a forward step toward the actual use of 1T' MoS$_2$.

**Discussion**

In summary, employing FeCoNi-LDH-NWAs as the self-sacrificing template and precursor, we have successfully synthesized FeCoNi-HNTAs using Ni foam as the substrate through a facile solvothermal method as a highly active and stable bifunctional electrocatalytst for overall water splitting in alkaline electrolyte. Notably, this complex polymetallic sulfides system

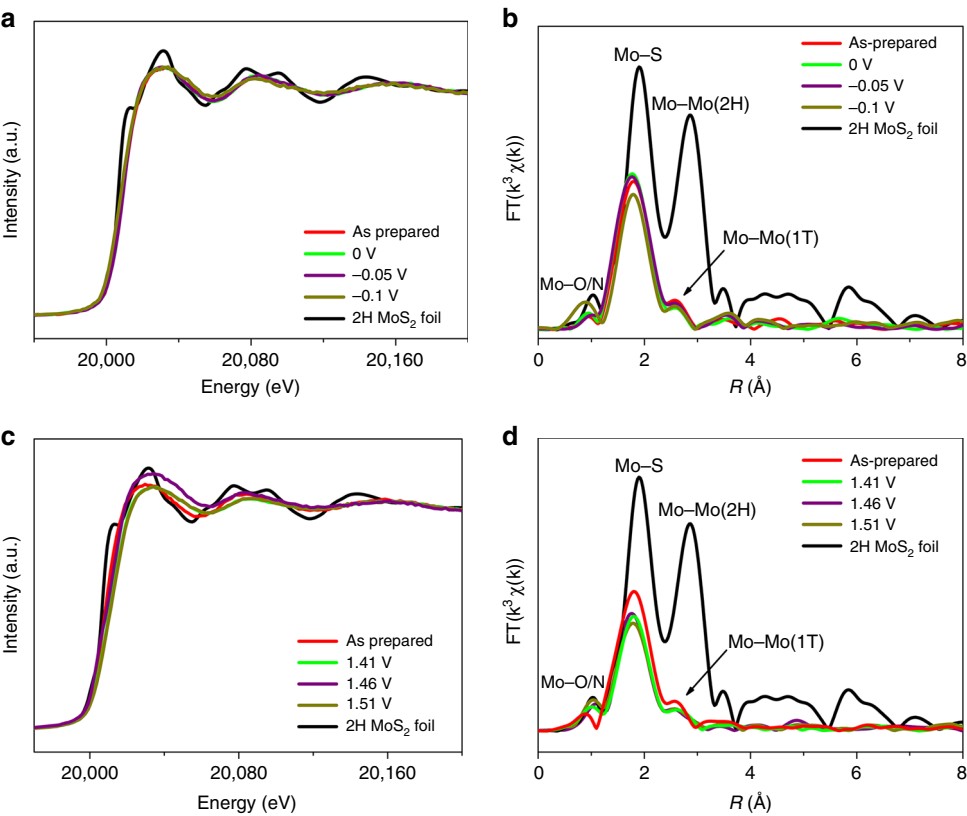

**Fig. 7** In situ EXAFS characterization showing phase stability of 1T' MoS$_2$. **a**, **c**, and **b**, **d**, The normalized in situ XANES spectra and the corresponding k$^3$-weighted FT profiles in R-space from EXAFS at Mo K-edge collected on as-synthesized FeCoNi-HNTAs in the processes of potentiostatic HER and OER measurements with different voltages (0, −0.05, and −0.1 V vs. RHE for HER and 1.41, 1.46, and 1.51 V vs. RHE for OER), respectively

leverages the multiple advantages of chemical compositions, geometric structural features, and substrate characteristics to demonstrate remarkably low overpotentials and long-term durability for electrolysis. Our comprehensive experimental explorations exhibit the compositional origins of the excellent electrocatalytic performance, which are proven to be the synergistic effects among Fe, Co, and Ni ions as well as proliferated catalytic active sites and superb conductivity of 1T' $MoS_2$. On the other hand, under-electrolyte superaerophobicity facilitates the disengagement of as-formed gas bubbles from electrode surface with ease, contributing to a significant boost in water splitting effectiveness. It is believed that this system-optimized hybrid electrode holds great promise for large-scale production and practical utilizations, which opens up a fascinating way to create high-efficiency electrodes for energy conversion technologies.

## Methods

**Synthesis of FeCoNi-LDH-NWAs**. In a typical synthesis, 121.2 mg $Fe(NO_3)_3 \cdot 9H_2O$, 436.5 mg $Co(NO_3)_2 \cdot 6H_2O$, 450 mg $(NH_2)_2CO$, and 222 mg $NH_4F$ were dissolved into 40 ml ultrapure water and moved into a 45 ml teflon-lined autoclave with steel shell. A piece of Ni foam ($1 \times 3$ cm, thickness: 1.5 mm, bulk density: 0.3 g/cm$^3$) was cleaned ultrasonically first with ethanol (20 ml), then with 3 M HCl aqueous solution (20 ml) for 15 min, and washed subsequently with deionized water for three times. The as-prepared Ni foam above was immersed into the above autoclave and exactly stuck at the center of the autoclave vertically. The autoclave was sealed and then heated at 120 °C for 10 h. The obtained electrode was cleaned with ethanol three times and dried in vacuum at room temperature for further characterizations and next step reaction. The synthetic methods of NiCo-LDH-NWAs and NiFe-LDH-NSAs were the same as that of FeCoNi-LDH-NWAs, except that no adding iron source for NiCo-LDH-NWAs and no adding cobalt source as well as increasing amount of iron source to 606 mg for NiFe-LDH-NSAs, respectively. When focusing on the details of reaction process, fluoride ions were added into the reaction solvent to coordinate with the metal ions, resulting in slow release of the metal ions and promotion of nucleus formation on the substrate. Then the released metal ions combined with $CO_3^{2-}$ and $OH^-$ generated from the hydrolysis of urea forming polymetal carbonate hydroxides.

**Synthesis of FeCoNi-HNTAs**. First, 15 mg of $(NH_4)_2MoS_4$ were dissolved in 38 ml of N,N-dimethylformamide (DMF) and stirred for 10 min to make solution homogeneous. Then 2 ml HZH (volume fraction, 85%) was introduced dropwise into the foregoing solution with vigorously stirring. The mixed solution for growing hybrid nanotubes was transferred into a 45 ml teflon-lined autoclave with steel shell and FeCoNi-LDH-NWAs was submerged into the above solution and exactly stuck at the direction of autoclave diameter vertically. The autoclave was sealed and then heated at 200 °C for 10 h. When the system was cooled down, this obtained electrode was cleaned with ethanol three times and dried in vacuum at room temperature for further characterizations. The synthetic methods of CoNi-HNTAs and NiFe-HNSAs were the same as that of FeCoNi-HNTAs, except that the precursor of FeCoNi-LDH-NWAs was replaced by NiCo-LDH-NWAs and NiFe-LDH-NSAs, respectively.

**Synthesis of $MoS_2$/Ni foam**. The synthetic method of $MoS_2$/Ni foam was the same as that of the FeCoNi-HNTAs, except that the precursor of FeCoNi-LDH-NWAs was replaced by the cleaned Ni foam.

**Synthesis of FeCoNiS-NTAs**. The synthetic method of FeCoNiS-NTAs was the same as that of the FeCoNi-HNTAs, except that the precursor of $(NH_4)_2MoS_4$ was replaced by 20 mg of TAA.

**Material characterizations**. The morphologies and structures of as-prepared samples were characterized by field-emission scanning electron microscopy, transmission electron microscopy, high-resolution transmission electron microscopy, and scanning transmission electron microscopy (FESEM: Hitachi, SU8010, 10 kV; TEM: Hitachi, H7700, 100 kV; HRTEM and STEM: JEOL, JEM-2100, 200 kV and FEI, Tecnai G2 F20, 200 kV). Atomic resolution aberration-corrected scanning transmission electron microscopy (AC-STEM) images were carried out by a JEOL ARM200F at 200 kV with double hexapole Cs correctors andcold field emission gun. Energy-disperse X-ray spectra (EDX) and elemental mapping spectra were performed with the same instruments as HRTEM. X-ray diffraction (XRD) patterns were measured by using a Bruker D8 Advance X-ray diffractometer equipped with Cu Kα radiation ($\lambda = 1.5418$ Å). X-ray photoelectron spectra (XPS) were performed by using a PHI Quantera SXM spectrometer with monochromatic Al Kα X-ray sources (1486.6 eV) at 2.0 kV and 20 mA. The wetting ability of surface of the electrode was characterized by surveying the contact angles of electrolyte, 1 M KOH solution. In the typical measurement, a 3 μl droplet of 1 M

KOH solution was dropped on surface of the electrode and the contact angle was measured by a Dataphysics OCA20 system at room temperature in ambient air. The contact angles of gas bubbles under electrolyte were tested by the method of captive bubble using Dataphysics OCA20 system. The observed equilibrium contact angles were defined as the pinned bubbles with liquid around on surface of the electrode, and the interface of liquid/gas contacts with the interface of solid/liquid across the three phase contact interfaces. The volume of the measured bubble was taken around 3 μl. When measuring the advancing angle and receding angle, it is needed to increase and then decrease a constant volume (ca. 2 μl) of the measured bubbles. Each contact angle measurement was repeated more than 5 times. The images of hydrogen and oxygen bubbles release were obtained by a high-speed charge-coupled device camera (i-SPEED, OLYMPUS) equipped with a microscope (SZ-CTC, OLYMPUS). A fiber optic illuminator system (Multi-Position, Nikon) provides the illumination. The ex situ and in situ synchrotron radiation-based extended X-ray absorption fine structure (EXAFS) measurements at Mo K-edge were carried out at the beamline 14W1 in 1W1B station in Beijing Synchrotron Radiation Facility (BSRF). A double-crystal Si (111) monochromator monochromatized the X-ray. The energy calibration was performed by using a Mo metal foil at Mo K-edge. Higher harmonics was removed by detuning the monochromator. The obtained data were calculated and processed based on the Win-XAS3.1 program[48]. Phase-shift functions and theoretical amplitudes were calculated by the FEFF8.2 code and the parameters of crystal structure of the standard 2H $MoS_2$ foil[49]. The synchrotron radiation-based soft X-ray absorption spectroscopy (sXAS) tests were carried out in 4B7B station in Beijing Synchrotron Radiation Facility (BSRF).

**Electrocatalytic measurements**. The electrocatalytic measurements were performed at room temperature with a Princeton PASTAT4000 and a Chenhua CHI660e instrument in 1 M KOH media. The HER and OER electrocatalytic processes were investigated in a typical three-electrode configuration. A Ag/AgCl electrode with saturated KCl–AgCl solution and a graphite rod were utilized as the reference and counter electrodes, respectively. The as-prepared electrodes supported on Ni foam were utilized as the working electrode. In this three-electrode configuration, the reversible hydrogen electrode (RHE) calibration was carried out in the $H_2$-saturated 1 M KOH solution using a Pt sheet as the working electrode. A single cycle of cyclic voltammetry was recorded at a scan rate of 1 mV s$^{-1}$ and the average value of the two potentials where the current crossed line of zero value was considered as the potential of the calibration. In 1 M KOH electrolyte, $E(RHE) = E$ (SCE) $+ 1.012$ V. Before the measurements, the working electrode was tailored as a square piece with the working surface area of about 0.9 cm$^2$ and immersed in the applied electrolyte for 12 h in vacuum to activate the electrode. For comparison, commercial Pt/C (20%, loaded on Ni foam with 1 mg cm$^{-2}$) and $IrO_2$/C (20%, loaded on Ni foam with 1 mg cm$^{-2}$) were deployed for HER and OER catalysis. Besides, overall water splitting was measured in a typical two-electrode system utilizing the as-prepared electrodes as cathode and anode simultaneously. For HER and OER processes, the polarization curves and Tafel plots were characterized in the selected potential ranges at a scan rate of 1 mV s$^{-1}$. The Nyquist plots of electrochemical impedance spectroscopy (EIS) was surveyed in the frequency range of 10 kHz–100 mHz at the open circuit voltage of HER and OER, respectively. The stability measurements ($i$–$t$) were recorded for 80 h at a constant applied potential of $-0.326$ V vs. RHE for HER and 1.796 V vs. RHE for OER, respectively. The cyclic voltammetry was carried out to assess electrochemical double-layer capacitance ($C_{dl}$) at no faradic processes six times at six different scan rates (1, 2, 3, 4, 5, 6 mV s$^{-1}$). The obtained current densities at the selected potential have the linear relationship with the scan rates and the slopes of fitting curves were considered as the $C_{dl}$. We applied the specific capacitance (20–60 μF cm$^{-2}$) of 40 μF cm$^{-2}$ here to calculate the ECSA. According to the Eq. (1)[50]

$$ECSA = \frac{C_{dl}}{40\,\mu F/cm^2}\ cm^2_{ECSA}. \tag{1}$$

the ECSA values were roughly obtained. The periodic galvanic pulses was achieved to prove the structural recoverability and stability of the material between two different current density of 100 mA cm$^{-2}$ and 200 mA cm$^{-2}$ with an iteration period of 2000 s. Moreover, gas chromatography (Shimadzu, GC-2014C) was employed to determine the experimentally evolved amount of $H_2$ and $O_2$ during 120 min in electrocatalytic processes for the Faradaic efficiency measurements. It was hypothesized for the theoretical values that the whole current transfers into the gas production during the reaction. We also calculate the theoretical quantity of $H_2$ and $O_2$ by using the Faraday law based on an $i$–$t$ curve at the constant current density of 20 mA cm$^{-2}$ for 120 min, which demonstrated that the transformation of 96485.4 C charge causes 1 equivalent of reaction. As for overall water splitting, the polarization curves were performed at a scan rate of 1 mV s$^{-1}$. Chronoamperometric curves of FeCoNi-HNTAs and commercial $IrO_2$/C-Pt/C couple electrode were measured at the constant cell voltages of 1.59 and 1.8 V respectively.

**Data availability**. The authors make a statement that the data presented by this article are available from the corresponding author on reasonable requests.

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

## Acknowledgements

This work was supported by China Ministry of Science and Technology under Contract of 2017YFA0700101 and 2016YFA0202801 and National Natural Science Foundation of China (21431003, 21521091). We thank Dr. Lirong Zheng for providing the EXAFS tests in 1W1B station and Dr. Shuhu Liu, Dr. Jiaou Wang for offering sXAS tests in 4B7B station in Beijing Synchrotron Radiation Facility (BSRF).

## Author contributions

X.W. supervised this study. X.W. and H.L. conceived the idea. H.L. planned and carried out the experiments, collected and analyzed the experimental data. S.C. and L.S. analyzed the ex situ and in situ EXAFS data. Y.Z. and X.S. collected the data of adhesive force and contact angle measurements. Q.H.Z. and L.G. performed AC-STEM characterizations. H.

L., X.J., Q.Z., and X.W. co-wrote the manuscript. All authors approve the final version of this article.

## Additional information

**Competing interests:** The authors declare no competing interests.

