## [Peer Review File · Nature Communications]

Reviewers' comments:

Reviewer #1 (Remarks to the Author):

Water electrolyser can generate the high purity hydrogen gas with precious metal catalysts. The non-precious metal catalysts (NPMC) with high columbic efficiency are important for the wide application of device. The authors developed one of NPMCs, which can be efficiently used for both HER and OER. The materials were well characterized with the different techniques, and the logic was reasonable. In my opinion, it is the first time to report one of such material with the small overpotential gap (1.43 V) at 10 mA cm⁻² for water electrolyser. The study is attractive for the reviewer, and it can be accepted for publication after considering the following recommended revisions.

1. Schematic illustration for the synthesis of hybrid electrocatalyst is important for readers to easily follow their strategies, which is strongly suggested for this matter.
2. The reviewer did not catch the synergetic effect of MoS and FeCoNi-HNTAs. It is the hybrid material, composite materials, or other?
3. The synergetic effect of Fe, Co and Ni for OER performance should be carefully presented, although it is really hard for the community to discuss the active site.

Reviewer #2 (Remarks to the Author):

This paper by Li et al reported the design of composite nanotube arrays as well as their performance as overall water splitting electrocatalysts. Water splitting is regarded as one of the potential candidates of clean energy by providing hydrogen sources. However the current technique faces the problem of noble metal catalysts, high energy consuming and poor long-term stability, etc. In this paper authors demonstrated the design of non-noble metal catalysts electrode by combining electrocatalytically active 1-T MoS₂ and metal sulfide. Meanwhile the nanotube arrays provide a superaerophobic surface, which makes it easier for gas bubbles to release from surfaces. It is a clever design and the performance is quite impressive by achieving the record low overpotential for overall water splitting reported up to date. I would like to recommend its publication after the following minor revisions:

1. The current electrode is composed of Fe, Co and Ni sulfide. The detailed role of each element may be further emphasized.
2. Authors provided impressive atomic-level HRTEM of the samples. It would be helpful if they provide corresponding crystal structures of the compounds in these figures.
3. The contrast of insets in Figure 1i and j are not satisfactory. Authors may use different colors or provide them as individual ones.

Reviewer #3 (Remarks to the Author):

Summary

The authors have developed a water splitting catalyst consisting of 1T' MoS₂ and Fe, Co, Ni-based sulphides in a nanotube-array configuration. This bifunctional catalyst demonstrated a current density

of 10 mA/cm² at overpotentials of 58 mV and 184 mV for HER and OER, respectively. A total cell potential of 1.429 V produced a current density of 10 mA/cm² with the catalyst acting as both anode and cathode. The authors claim that this performance is due to proliferated catalytic sites obtained by the nanotube-array architecture and high conductivity of 1T' MoS₂. A synergistic effect between Fe, Co, and Ni ions along with superaerophobicity of the catalyst surface also played a role in the high performance of the catalyst. Additionally, the catalyst proved to be stable over 80 h of operation at high current densities (200 mA/cm²).

General comments

The performance of the catalyst is impressive. The overpotentials required to reach 10 mA/cm² are exceptionally low, especially for a bifunctional catalyst. The catalyst also exhibits excellent stability over an extended (for this field at least) period of time. The authors demonstrate a systematic approach to designing their catalyst, and characterization of their materials is thorough. Control experiments are well utilized.

The author's claims are well supported by their data. Electrochemical surface area (ECSA) experiments along with scanning electron microscopy (SEM) data support the claim of a high number of catalytic sites compared to the control materials. The high conductivity of 1T' MoS₂ is supported by impedance spectroscopy. Synergy between Fe, Co, and Ni ions is demonstrated using soft X-ray absorption spectroscopy (sXAS) on the catalyst and on multiple control materials. Fast reaction kinetics are backed up using polarization curves to demonstrate HER and OER electrocatalytic performance. Stability of the catalyst is verified both electrochemically and with X-ray absorption (EXAFS) experiments.

Overall, the science in this paper is of high quality and the results are impactful. The authors combine known knowledge of state of the art electrodes: 1T' MoS₂; Fe, Co, Ni based sulfides; and nanotube-arrays to produce a novel water splitting electrocatalyst. The authors do a good job of building on their previous work (Nat Comm 2017) where they combine amorphous NiCo with 1T' MoS₂ to obtain overpotentials of 70 mV for HER and 235 mV for OER at 10 mA/cm².

We do have several comments below that should be addressed.

The paper also contains numerous grammatical errors that detract from the manuscript.

When the catalyst is used as both anode and cathode the total cell potential is 1.429 V which gives a total overpotential for both HER and OER of 0.199 V. This is smaller than the total overpotential when the catalyst is evaluated separately for OER and HER (0.242 V). Do the authors have an explanation for this?

Figure S6: The XRD pattern of FeCoNi-HNTAs does not appear to display the prominent peaks of MoS₂ around a 2 theta of 15, 40, and 50 degrees. There is also a peak in the FeCoNi-HNTAs pattern around 23 degrees that does not appear in either reference pattern. Do the authors have an explanation for these peaks?

Figure 2c-f: The signal-to-noise ratio in Figure 2d is very low and it's difficult to accept the fitting of this data. It would be preferable to obtain better quality XPS data for Fe. Figures 2c,e,f show the raw data as points rather than lines (Figure 2d). Having a consistent format for the XPS data would be helpful.

Figures 3, S10: These figures are confusing with the intermixing of roman and arabic numerals. I am not clear if insets I-III correspond to the 1-3 labels on the plot and if so, is there any significance to

the labeling of insets IV-VI? It does not help that the caption does not display the roman numerals correctly. Better clarification on what refers to what is needed here.

Figure 4a,b: Figure 5a shows a dotted horizontal line at 10 mA/cm². That line would be useful to have in this figure.

Figure 4c,g: These plots show different Cdl values for the HER and OER catalysts. Since these are the same material, it should be expected that they have the same Cdl. Figure S15 shows that the measurements were taken over different ranges potentials for the HER and OER catalysts which may play a part in the different Cdl values. Or is there a large variance in surface area between samples? Some discussion here would be beneficial.

Figure 4h: The performance of the OER catalyst seems to steadily improve over time. An improvement is also seen for overall cell performance Figure 5b. Do the authors have a possible reason for this improvement in performance?

Figure S18: Including the XRD pattern for the material before electrochemical operation would allow an easier comparison of the material before and after operation.

Supplementary Table 2: The column width needs to be adjusted as words are being cut off.

Reply to Referee 1 and revisions made accordingly:

Water electrolyser can generate the high purity hydrogen gas with precious metal catalysts. The non-precious metal catalysts (NPMC) with high columbic efficiency are important for the wide application of device. The authors developed one of NPMCs, which can be efficiently used for both HER and OER. The materials were well characterized with the different techniques, and the logic was reasonable. In my opinion, it is the first time to report one of such material with the small overpotential gap (1.43 V) at 10 mAcm^{-2} for water electrolyser. The study is attractive for the reviewer, and it can be accepted for publication after considering the following recommended revisions.

We are grateful to the reviewer's comments. We have made revisions according to each comment, as summarized below.

1. Schematic illustration for the synthesis of hybrid electrocatalyst is important for readers to easily follow their strategies, which is strongly suggested for this matter.

Reply: Thanks for the comments and suggestions from the reviewer. We have added the schematic illustration as new Fig. 1a in the revised manuscript to exhibit the synthetic process of FeCoNi-HNTAs, which is intelligible for readers to catch the synthetic strategy. Supplementary Fig. 4 shows the FESEM and TEM images of the products at different reaction stages corresponding to Fig. 1a.

Scheme R1. Schematic illustration showing the synthesis of FeCoNi-HNTAs with crystalline structures of $1\text{T}' \text{MoS}_2$ and $(\text{Co, Fe, Ni})_9\text{S}_8$ from the corresponding precursor of FeCoNi-LDH-NWAs.

Revision made: We have added Scheme R1 to the revised manuscript as Fig. 1a and removed the schematic representations in Supplementary Fig. 4.

2. The reviewer did not catch the synergetic effect of MoS and FeCoNi-HNTAs. It is the hybrid material, composite materials, or other?

Reply: Thanks for the comments from the reviewer. According to the characterizations shown in the revised manuscript, such as HRTEM and AC-STEM (Fig. 1f-h), XRD (Supplementary Fig. 9), EXAFS (Fig. 2a,b), and XPS (Fig. 2c-f), we can confirm the compositions of FeCoNi-HNTAs are $1\text{T}' \text{MoS}_2$ and $(\text{Co, Fe, Ni})_9\text{S}_8$. The hybrid material is considered as the material involves interacting multi-compositions without crystal lattice connections while the composite material is deemed to be the materials comprises noninteracting multi-compositions with physical mixture. Therefore, we

think FeCoNi-HNTAs are the hybrid materials. Firstly, as shown in Fig. 1d in the revised manuscript, seven elements distribute uniformly in the nanotube, which means both the two compositions coexist and uniformly distribute in FeCoNi-HNTAs. Secondly, based on Fig. 1f and Supplementary Fig. 19c,d, there are no crystal lattice boundaries between the two compositions that can be distinguished. Meanwhile, as shown in Fig. 2b, Mo atoms do not bound to other metal atoms. Therefore, FeCoNi-HNTAs are not heterostructures or single component material.

Figure R1. High-resolution XPS spectra comparison of Mo 3d regions between FeCoNi-HNTAs and MoS₂/Ni Foam.

Figure R2. a, FESEM images of FeCoNiS-NTAs. Scale bar: 1 μm. **b,** Gas bubble adhesive force measurements of FeCoNiS-NTAs showing no adhesive force to the bubble on the surface. The insets 1-3 show the bubble adhesive force measurement process and no distinct bubble deformation can be observed in this process, further illustrating the under-water superaerophobicity. For the

bubble contact angle under water (inset 4), it was measured to be $152.9^\circ \pm 3.4^\circ$. Inset 5 display the superhydrophilicity of the electrode because KOH solution droplets could not be captured. **c,d**, Digital photos showing the as-formed bubbles releasing behaviors on the surface of FeCoNiS-NTAs for OER and HER respectively. Scale bar: 2 mm.

Figure R3. a,f, Polarization curves for HER and OER measured at a scan rate of 1 mVs⁻¹ in 1 M KOH solution. **b,g,** Tafel plots for HER and OER measured at a scan rate of 1 mVs⁻¹. **c,h,** The fitting plots showing C_{dl} for HER and OER. **d,i,** EIS Nyquist plots at the open circuit voltage for HER and OER respectively. Insets are detailed illustrations for showing the much smaller resistance of FeCoNi-HNTAs. **e,j,** Cyclic voltammetry for HER and OER respectively at the scan rates of 1, 2, 3, 4, 5, 6 mVs⁻¹ in the range of no Faradaic processes for measuring C_{dl} of FeCoNiS-NTAs.

Furthermore, we compare the XPS data of Mo 3d regions between FeCoNi-HNTAs and MoS₂/Ni Foam samples (Figure R1). The obvious downshift about 0.8 eV of the peak positions of FeCoNi-HNTAs compared to that of MoS₂/Ni Foam can be observed, which elucidates the charge transfer from Fe, Co, Ni-based sulfides to MoS₂. This interaction proves the hybrid material nature of FeCoNi-HNTAs and demonstrates the synergetic effect of Fe, Co, Ni-based sulfides and MoS₂. Moreover, we added some control experiments to verify the synergetic effect of Fe, Co, Ni-based sulfides and MoS₂ in electrocatalytic performance. We introduced FeCoNiS-NTAs to highlight the performance superiority of the hybrid material, FeCoNi-HNTAs, for water splitting electrocatalysis. The thioacetamide was used to sulfurate FeCoNi-LDH-NWAs precursor instead of (NH₄)₂MoS₄, resulting in Fe,Co,Ni-based sulfides porous nanotube arrays (Figure R2a). The surface of electrode is also superaerophobic under electrolyte and superhydrophilic according to Figure R2b and thus the as-formed bubbles can release from the electrode surface with ease (Figure R2c,d). HER, OER and OWS catalytic performances of FeCoNiS-NTAs were added into the revised manuscript and Supplementary Information. When making the comparisons with that of MoS₂/Ni Foam and FeCoNiS-NTAs, the catalytic performances of FeCoNi-HNTAs are much better (Figure R3). The results can clarify the synergetic effect between the two compositions.

Revision made: We have added Figure R1 to the revised manuscript as Supplementary Fig. 12. Figure R2 have been provided in the revised Supplementary Information as Supplementary Fig. 6b, 13d, 10g,f. Figure R3 have been added into the revised manuscript and Supplementary Information as Fig. 4a-c, e-g and Supplementary Fig. 16, 17g-h. The OWS polarization curve of FeCoNiS-NTAs has been added into Fig. 5a in the revised manuscript.

3. The synergetic effect of Fe, Co and Ni for OER performance should be carefully presented, although it is really hard for the community to discuss the active site.

Reply: Thanks for the suggestions from the reviewer. According to Supplementary Fig. 15 in the revised Supplementary Information, FeCoNi-HNTAs demonstrate superior electrocatalytic performance for OER compared with CoNi-HNTAs and NiFe-HNSAs, which means the three metals are all indispensable for manufacturing high-performance electrode for OER catalysis. In the main text of the revised manuscript, we mainly used sXAS on the L₃ regions at Fe, Co and Ni L-edge to demonstrate the synergetic effect of Fe, Co and Ni for OER electrocatalysis. Firstly, for Fe ions, Fe³⁺ has been proved as the highly active sites for OER because the intermediate of OER has quite low Gibbs free energy of adsorption on Fe sites (*J. Am. Chem. Soc.* 2015, 137, 1305.). Meanwhile, appropriate doping quantity of Fe are beneficial to lower the overpotentials and enhance the intrinsic activity of Ni-based compounds (oxyhydroxide, sulfide, etc.) for OER (*J. Am. Chem. Soc.* 2013, 135,

12329; *J. Mater. Chem. A* 2016, 4, 13499). In our case, compared with NiFe-HNSAs, FeCoNi-HNTAs possess larger amount of Fe³⁺ according to the sXAS at Fe L₃ edge (Fig. 2g), indicating Co ions can adjust the electronic structure of Fe cations to provide the high valence state. As shown in Supplementary Fig. 8, the atomic ratio of Fe in the three metals is about 11.24%, which tends to exert a partial-charge-transfer activation effect on Ni and thus improve the catalytic activity of OER (*J. Am. Chem. Soc.* 2014, 136, 6744.). Secondly, it has been reported that Co³⁺ ions located at octahedral sites are the active centers for OER rather than the Co²⁺ ions at tetrahedral sites (*Angew. Chem. Int. Ed. Engl.* 2015, 127, 7507; *Energy Environ. Sci.* 2013, 6, 926.). At the meantime, the interaction between Fe and Co ions is considered to improve the intrinsic activity for OER and affects chemical and structural stability (*J. Am. Chem. Soc.* 2015, 137, 3638.) Based on the sXAS at Co L₃ edge in the revised manuscript (Fig. 2h), FeCoNi-HNTAs have more Co³⁺ ions located at octahedral sites than CoNi-HNTAs, which demonstrates that the interaction between Fe and Co ions can tune the crystal-field coordination of Co ions and thus enhance the catalytic activity. Based on the above two parts, the synergetic effect of Fe and Co ions in our hybrid system for OER performance are perspicuous. Thirdly, previous studies (*J. Am. Chem. Soc.* 2015, 137, 1305; *J. Phys. Chem. C* 2015, 119, 27228.) have been demonstrated that Ni³⁺ ions have higher OER catalytic activity in contrast to Ni²⁺ ions. And Fe and Co respective incorporations for the OER performance enhancement of Ni-based compounds have been investigated (*Nano Energy* 2016, 27, 526; *ACS Nano*, 2017, 11, 9550.). We made a comparison among the sXAS at Ni L₃ edge of FeCoNi-HNTAs, CoNi-HNTAs and NiFe-HNSAs (Fig. 2i), and the obvious increase of Ni³⁺ quantity could be observed on that of FeCoNi-HNTAs. Therefore, the interaction between Fe, Co and Ni in our hybrid system adjusted the local electronic structure of Ni cations and thus improve the OER performance. In summary, the synergetic effect of Fe, Co and Ni have a great significance on OER performance enhancement.

Revision made: We have added some illustrations about the synergetic effect of Fe, Co and Ni for OER performance.

We truly thank the reviewer for the insightful comments and kind suggestions! The reply for each question/comment is expected to reach the high criteria.

Reply to Referee 2 and revisions made accordingly:

This paper by Li et al reported the design of composite nanotube arrays as well as their performance as overall water splitting electrocatalysts. Water splitting is regarded as one of the potential candidates of clean energy by providing hydrogen sources. However the current technique faces the problem of noble metal catalysts, high energy consuming and poor long-term stability, etc. In this paper authors demonstrated the design of non-noble metal catalysts electrode by combining electro-catalytically active 1-T MoS₂ and metal sulfide. Meanwhile the nanotube arrays provide a superaerophobic surface, which makes it easier for gas bubbles to release from surfaces. It is a clever design and the performance is quite impressive by achieving the record low overpotential for overall water splitting reported up to date. I would like to recommend its publication after the following minor revisions:

We are grateful to the reviewer's comments. We have made revisions according to each comment, as summarized below.

1. The current electrode is composed of Fe, Co and Ni sulfide. The detailed role of each element may be further emphasized.

Reply: Thanks for the suggestions from the reviewer. According to Supplementary Fig. 15 in the revised Supplementary information, FeCoNi-HNTAs demonstrate superior electrocatalytic performances for OER compared with CoNi-HNTAs and NiFe-HNSAs, which means the three metals are all indispensable for manufacturing high-performance electrode for OER catalysis. In the main text of the revised manuscript, we mainly used sXAS on the L₃ regions at Fe, Co and Ni L-edge to demonstrate the roles of Fe, Co and Ni in OER electrocatalysis. Firstly, for Fe ions, Fe³⁺ has been proved as the highly active sites for OER because the intermediate of OER has quite low Gibbs free energy of adsorption on Fe sites (*J. Am. Chem. Soc.* 2015, 137, 1305.). Meanwhile, appropriate doping quantity of Fe are beneficial to lower the overpotentials and enhance the catalytic activity of Ni-based compounds (oxyhydroxide, sulfides, etc.) for OER (*J. Am. Chem. Soc.* 2013, 135, 12329; *J. Mater. Chem. A* 2016, 4, 13499). In our case, compared with NiFe-HNSAs, FeCoNi-HNTAs possess larger amount of Fe³⁺ according to the sXAS at Fe L₃ edge (Fig. 2g), indicating Co ions can adjust the electronic state of Fe to provide the high valence state. As shown in Supplementary Fig. 8, the atomic ratio of Fe in the three metals is about 11.24%, which tends to exert a partial-charge-transfer activation effect on Ni and thus improve the catalytic activity of OER (*J. Am. Chem. Soc.* 2014, 136, 6744.). Secondly, it has been reported that Co³⁺ ions located at octahedral sites are the active centers for OER rather than the Co²⁺ ions at tetrahedral sites (*Angew. Chem. Int. Ed. Engl.* 2015, 127, 7507; *Energy Environ. Sci.* 2013, 6, 926.). At the meantime, the interaction between Fe and Co ions is considered to improve the intrinsic activity for OER and affects chemical and structural stability (*J. Am. Chem. Soc.* 2015, 137, 3638.). Based on the sXAS at Co L₃ edge in the revised manuscript (Fig. 2h), FeCoNi-HNTAs have more Co³⁺ ions located at octahedral sites than CoNi-HNTAs, which demonstrates that the interaction between Fe and Co ions can tune the crystal-field coordination of Co ions and thus enhance the catalytic activity. Thirdly, previous studies (*J. Am. Chem. Soc.* 2015, 137, 1305; *J. Phys. Chem. C* 2015, 119, 27228.) have been demonstrated that Ni³⁺ ions have higher OER catalytic activity in contrast to Ni²⁺ ions. And Fe and Co respective incorporations for the OER performance enhancement of Ni-based compounds have been investigated (*Nano Energy* 2016, 27, 526; *ACS Nano*, 2017, 11, 9550.). We made a comparison among the sXAS at Ni L₃ edge of FeCoNi-HNTAs, CoNi-HNTAs and NiFe-HNSAs (Fig. 2i), and the obvious increase of Ni³⁺ quantity could be observed on that of FeCoNi-HNTAs. Therefore, the interaction between Fe, Co and Ni in our hybrid system adjusted the local electronic structure of Ni cations and thus improve the OER performance. In summary, the Fe, Co and Ni ions play extremely significant roles in OER performance enhancement.

Revision made: We have added some illustrations about the roles of Fe, Co and Ni in OER electrocatalysis.

2. Authors provided impressive atomic-level HRTEM of the samples. It would be helpful if they provide corresponding crystal structures of the compounds in these figures.

Reply: Thanks for the suggestions from the reviewer. We have provided the crystal structures of 1T' MoS₂ and (Co,Fe,Ni)₉S₈ in Figure R4.

Figure R4. a,b, AC-STEM images of 1T' MoS₂ and (Co,Fe,Ni)₉S₈ in FeCoNi-HNTAs respectively. The upper image of inset of **a** displays the Mo clusters with tetragonal symmetry marked by short dash line square in **a**. The corresponding FFT pattern of (Co,Fe,Ni)₉S₈ is shown as the upper image of inset of **b**. The lower images of the insets of **a** and **b** are the corresponding crystal structures, where yellow, cyan and blue balls represent S, Mo and Co (or Fe, Ni) atoms.

Revision made: We have added Figure R4 into the revised manuscript as Fig. 1g and 1f.

3. The contrast of insets in Figure 1i and j are not satisfactory. Authors may use different colors or provide them as individual ones.

Reply: Thanks for the suggestions from the reviewer. We have used different colors to show the insets in Fig. 1i-k and Supplementary Fig. 10, which have obvious contrast.

Figure R5. a-c Digital photos demonstrating the bubble releasing behaviors on the surface of FeCoNi-HNTAs, FeCoNi-LDH-NWAs and bare Ni foam for HER. The insets are the corresponding statistics of size distribution of releasing bubbles. Scale bars: 2 mm.

Revision made: We have added Figure R5 into the revised manuscript as Fig. 1i-k and also used different colors to exhibit the insets in Supplementary Fig. 10.

We truly thank the reviewer for the insightful comments and kind suggestions! The reply for each question/comment is expected to reach the high criteria.

Reply to Referee 3 and revisions made accordingly:

Summary

The authors have developed a water splitting catalyst consisting of 1T' MoS₂ and Fe, Co, Ni-based sulphides in a nanotube-array configuration. This bifunctional catalyst demonstrated a current density of 10 mA/cm² at overpotentials of 58 mV and 184 mV for HER and OER, respectively. A total cell potential of 1.429 V produced a current density of 10 mA/cm² with the catalyst acting as both anode and cathode. The authors claim that this performance is due to proliferated catalytic sites obtained by the nanotube-array architecture and high conductivity of 1T' MoS₂. A synergistic effect between Fe, Co, and Ni ions along with superaerophobicity of the catalyst surface also played a role in the high performance of the catalyst. Additionally, the catalyst proved to be stable over 80 h of operation at high current densities (200 mA/cm²).

General comments

The performance of the catalyst is impressive. The overpotentials required to reach 10 mA/cm² are exceptionally low, especially for a bifunctional catalyst. The catalyst also exhibits excellent stability over an extended (for this field at least) period of time. The authors demonstrate a systematic approach to designing their catalyst, and characterization of their materials is thorough. Control experiments are well utilized.

The author's claims are well supported by their data. Electrochemical surface area (ECSA) experiments along with scanning electron microscopy (SEM) data support the claim of a high number of catalytic sites compared to the control materials. The high conductivity of 1T' MoS₂ is supported by impedance spectroscopy. Synergy between Fe, Co, and Ni ions is demonstrated using soft X-ray absorption spectroscopy (sXAS) on the catalyst and on multiple control materials. Fast reaction kinetics are backed up using polarization curves to demonstrate HER and OER electrocatalytic performance. Stability of the catalyst is verified both electrochemically and with X-ray absorption (EXAFS) experiments.

Overall, the science in this paper is of high quality and the results are impactful. The authors combine known knowledge of state of the art electrodes: 1T' MoS₂; Fe, Co, Ni based sulfides; and nanotube-arrays to produce a novel water splitting electrocatalyst. The authors do a good job of building on their previous work (Nat Comm 2017) where they combine amorphous NiCo with 1T' MoS₂ to obtain overpotentials of 70 mV for HER and 235 mV for OER at 10 mA/cm².

We are grateful to the reviewer's comments. We have made revisions according to each comment, as summarized below.

We do have several comments below that should be addressed.

1. The paper also contains numerous grammatical errors that detract from the manuscript.

Reply: Thanks for the comments from the reviewer. We have corrected the grammatical and clerical errors in the revised manuscript and Supplementary Information.

2. When the catalyst is used as both anode and cathode the total cell potential is 1.429 V which gives a total overpotential for both HER and OER of 0.199 V. This is smaller than the total overpotential when the catalyst is evaluated separately for OER and HER (0.242 V). Do the authors have an explanation for this?

Reply: Thanks for the comments and suggestions from the reviewer. The measured potentials in HER and OER electrocatalysis have quite different significance compared with the measured voltage in overall water splitting system. For overall water splitting electrocatalysis, we used the two-electrode cell system to perform this measurement. The two electrodes are the same samples, the one as the working electrode and the other as both reference and counter electrodes. Therefore, the measured cell voltage represents the voltage difference between the two electrodes. For HER and OER electrocatalysis, we used the typical three-electrode cell system. The as-prepared samples are utilized as the working electrode, the Ag/AgCl electrode as the reference electrode and the graphite rod as the counter electrode. The Ag/AgCl electrode is a kind of standard electrode, which can be used as an indicator electrode. Therefore, the measured potentials in HER and OER systems represent the potentials of the working electrode versus the standard reference electrode. Moreover, the electrolyte between the working electrode and the reference electrode may produce solution resistance to affect the measured potential (or voltage). Our as-prepared electrodes have different shapes and sizes compared to the Ag/AgCl electrode so that the distances between the working electrode and the reference electrode are different in the two cell systems, which leads to the different solution resistances. So the total overpotential in overall water splitting system is different than that evaluated separately for OER and HER.

Revision made: We have changed all of the illustrations of overpotentials (η_{OWS}) in overall water splitting part of the revised manuscript to the cell voltage (E_{OWS}).

3. Figure S6: The XRD pattern of FeCoNi-HNTAs does not appear to display the prominent peaks of MoS₂ around a 2 theta of 15, 40, and 50 degrees. There is also a peak in the FeCoNi-HNTAs pattern around 23 degrees that does not appear in either reference pattern. Do the authors have an explanation for these peaks?

Reply: Thanks for the comments and suggestions from the reviewer. Compared to the substrate (Ni foam), the as-prepared MoS₂ has poorer crystallinity, which results in the weaker peak intensity in XRD pattern (the original one). We remeasured the XRD pattern of FeCoNi-HNTAs using an instrument with larger power so that the prominent peaks of MoS₂ can be detected. As shown in Supplementary Fig. 9 in the revised Supplementary Information, the XRD pattern of FeCoNi-HNTAs can match well with the standard MoS₂ peaks. Meanwhile, according to the as-measured EXAFS data, HRTEM and AC-STEM images, XPS spectrum and EDX elemental

mapping spectra, we can confirm the MoS_2 as one of the compositions in FeCoNi-HNTAs. As for the peak at around 23 degrees in the XRD pattern of FeCoNi-HNTAs, we made a comparison of XRD patterns of FeCoNi-HNTAs, NiFe-HNSAs and CoNi-HNTAs as shown in Figure R1 and demonstrated that this peak should be corresponding to Fe-involved compounds. Then we searched the database and found that this peak can match well with the standard Fe_2O_3 peak (JCPDS No. 39-0238). This result indicates that FeCoNi-HNTAs were oxidized slightly, which can be confirmed in Supplementary Fig. 8.

Figure R6. The comparison of XRD patterns of FeCoNi-HNTAs, NiFe-HNSAs and CoNi-HNTAs. The violet lines present the standard Fe_2O_3 (JCPDS No.39-0238) peaks.

Revision made: We have changed the pristine XRD patterns of FeCoNi-HNTAs in Supplementary Fig. 9, 20 to new ones.

4. Figure 2c-f: The signal-to-noise ratio in Figure 2d is very low and it's difficult to accept the fitting of this data. It would be preferable to obtain better quality XPS data for Fe. Figures 2c,e,f show the raw data as points rather than lines (Figure 2d). Having a consistent format for the XPS data would be helpful.

Reply: Thanks for the comments and suggestions from the reviewer. The XPS data of Fe have been remeasured and the quality of the spectrum is improved. Meanwhile, the raw data of Fig. 2d have been changed to points, which have a consistent format with that of Fig. 2c,e,f.

Figure R7. High-resolution XPS spectra of Fe 2p regions with fitting curves.

Revision made: We have added Figure R7 to the revised manuscript as Fig. 2d.

5. Figures 3, S10: These figures are confusing with the intermixing of roman and arabic numerals. I am not clear if insets I-III correspond to the 1-3 labels on the plot and if so, is there any significance to the labeling of insets IV-VI? It does not help that the caption does not display the roman numerals correctly. Better clarification on what refers to what is needed here.

Reply: Thanks for the comments and suggestions from the reviewer. We have changed all of the insets' labels of Fig. 3 and Supplementary Fig. 13 into arabic numerals. The insets 1-3 are corresponding to the 1-3 labels on the plot, where process 1 illustrates the electrode surface gets close to the air bubble, process 2 demonstrates the electrode surface leaves the air bubble and process 3 displays the electrode surface separates from the air bubble. The insets 4 show the bubble contact angles under electrolyte on the surface of the as-prepared electrodes, which demonstrate the superaerophobicity of the surface of as-prepared electrodes under electrolyte. The insets 5 exhibit the wetting ability of the electrodes, where KOH solution droplets can not be captured suggesting the surface superhydrophilicity of the electrodes. The two figures are used to clarify under-electrolyte superaerophobicity and superhydrophilicity of the electrodes, which are the reasons behind excellent electrocatalytic performance of gas evolution reaction.

Figure R8. a-h, Gas bubble adhesive force measurements of FeCoNi-HNTAs, FeCoNi-LDH-NWAs, MoS₂/Ni Foam, bare Ni foam, CoNi-HNTAs, NiFe-HNSAs, FeCoNi-HNWAs and FeCoNiS-NTAs respectively. The insets 1-3 show the bubble states in the corresponding measurement process of adhesive force where process 1 illustrates the electrode surface gets close to the air bubble, process 2 demonstrates the electrode surface leaves the air bubble and process 3 displays the electrode surface separates from the air bubble. The distinct bubble deformation observed in insets 2 of c and d further prove the adhesivity of MoS₂/Ni Foam and bare Ni foam to bubbles. For insets 4, the bubble contact angles under electrolyte were measured respectively. Insets 5 show the wetting ability of the electrodes in which KOH solution droplets can not be captured suggesting the surface superhydrophilicity of the employed electrodes.

Revision made: We have changed all of the insets' labels of Fig. 3 and Supplementary Fig. 13 into arabic numerals.

6. Figure 4a,b: Figure 5a shows a dotted horizontal line at 10 mA/cm². That line would be useful to have in this figure.

Reply: Thanks for the suggestions from the reviewer. The dotted horizontal lines at 10 mA/cm² were added in Fig. 4a,e and the corresponding overpotentials of FeCoNi-HNTAs for HER and OER were also shown.

Figure R8. a,b, Polarization curves for HER and OER measured at a scan rate of 1 mVs⁻¹ in 1 M KOH solution.

Revision made: We have added Figure R8 into the revised manuscript as Fig. 4a,e and provided the corresponding overpotentials of FeCoNi-HNTAs.

7. Figure 4c,g: These plots show different Cdl values for the HER and OER catalysts. Since these are the same material, it should be expected that they have the same Cdl. Figure S15 shows that the measurements were taken over different ranges potentials for the HER and OER catalysts which may play a part in the different Cdl values. Or is there a large variance in surface area between samples? Some discussion here would be beneficial.

Reply:

Figure R9. **a**, The fitting plots showing C_{dl} . **b-e**, Cyclic voltammetry at the scan rates of 1, 2, 3, 4, 5, 6 mV s^{-1} in the range of no Faradaic processes for measuring C_{dl} of FeCoNi-HNTAs, FeCoNiS-NTAs, MoS₂/Ni Foam and FeCoNi-LDH-NWAs.

Thanks for the comments and suggestions from the reviewer. It is indeed different potential ranges of cyclic voltammetry (CV) measurements that cause different C_{dl} values of HER and OER for one catalyst. The C_{dl} value is determined using CV measurements. The potential range represents a non-Faradaic current response, which is usually a 0.1 V window involving in open circuit potential (OCP) of the system. CV measurements are performed in quiescent electrolyte same as that of other electrocatalytic measurements by sweeping the potential across the non-Faradaic region from the more positive to more negative potential and back at different scan rates with the same interval. Then the different current densities at OCP are plotted with the different scan rates and the slope value of the fitting line is considered as the C_{dl} . Therefore, the potential range of CV depends on the obtained OCP. In our measurements, we measured OCP values of HER and OER for one catalyst after the measurements of CV polarization for activation, linear sweep voltammetry and Tafel scan. In this situation, the electrode has been polarized, which means it is in the catalytic state. We think that the

C_{dl} measured on the electrode in the catalytic state can demonstrate the actual double-layer capacitance more adequately. However, the different OCP values of HER and OER for one catalyst may be obtained in this situation. So the potential ranges of CV measurements are different for HER and OER and thus the C_{dl} values are different. These C_{dl} values we showed in Fig. 4 are convincing and reasonable because all of them are measured on the electrodes we synthesized that have been polarized under the same condition. In fact, the C_{dl} value is just used to estimate the ECSA roughly. The ECSA should be calculated by dividing the C_{dl} value by the specific capacitance of the sample. This specific capacitance is different in different electrolytes. We just used a general value of $40 \mu\text{Fcm}^{-2}$ in 1 M KOH to calculate ECSA and we preferentially emphasize the quantity order of C_{dl} values rather than the specific values in the main text of the revised manuscript. We also performed the CV measurements on the electrodes without polarization in the range of 0.1 V centered at OCP of the system and calculated the corresponding C_{dl} values as shown in Fig. R2, which kept the same for HER and OER. The double-layer current (i_c) is equal to the product of the scan rate (v) and double-layer capacitance (C_{dl}), which is shown as $i = vC_{dl}$. Therefore, the double-layer capacitance can be calculated based on the equation, $C_{dl} = d(\Delta j(\text{OCP}))/2dv$, where Δj represents the difference between the anodic and cathodic current densities at OCP (*J. Am. Chem. Soc.* 2015, 137, 4347; *Adv. Mater.* 2016, 28, 3785). The C_{dl} values we calculated in the measurements approach to the ones we showed in the revised manuscript for HER and OER, which demonstrates the data we provided in the revised manuscript are convincing and reasonable.

8. *Figure 4h: The performance of the OER catalyst seems to steadily improve over time. An improvement is also seen for overall cell performance Figure 5b. Do the authors have a possible reason for this improvement in performance?*

Reply: Thanks for the comments and suggestions from the reviewer. In fact, OER and overall water splitting performances of FeCoNi-HNTAs improve over time just in a small degree. For OER, the current density at 80 h increases just 8.14% compared to 200 mAcm^{-2} . And for overall water splitting, the current density at 100 h improve only 2.03% compared with 50 mAcm^{-2} . Here we used large current densities to demonstrate the OER and overall water splitting catalytic stability of FeCoNi-HNTAs. Under such large current densities, the consumption of water is so fast that the concentration of hydroxide ions increases over time. Although the set value of electrolysis potential is constant, the actual potential difference between the working electrode and the reference electrode increases, which results in larger current density. Moreover, the increase of ion concentration also contributes to the decrease of system resistance, which leads to larger current density. Therefore, the performance of OER and overall water splitting on FeCoNi-HNTAs steadily improve over time.

9. *Figure S18: Including the XRD pattern for the material before electrochemical operation would allow an easier comparison of the material before and after operation.*

Reply: Thanks for the suggestions from the reviewer. The pristine XRD pattern of FeCoNi-HNTAs has been added into Supplementary Fig. 20 for better comparison.

Figure R10. XRD pattern of FeCoNi-HNTAs after 1000 cycles of HER and OER. The pink lines present the standard MoS₂ (JCPDS No. 37-1492) peaks, the dark cyan lines demonstrate the standard (Co,Fe,Ni)₉S₈ (JCPDS No. 12-0723) peaks. The three peaks of the samples marked by stars represent nickel metal from the Ni foam.

Revision made: We have added Figure R10 into the revised Supplementary Information as Supplementary Fig. 20.

10. *Supplementary Table 2: The column width needs to be adjusted as words are being cut off.*

Reply: Thanks for the suggestions from the reviewer. The format of Supplementary Table 2 was adjusted to make the words together.

Revision made: We have adjusted the format of Supplementary Table 2.

We thank the reviewer for the insightful comments and kind suggestions! The reply for each question/comment is expected to reach the high criteria.

REVIEWERS' COMMENTS:

Reviewer #1 (Remarks to the Author):

The manuscript is acceptable with author's revision

Reviewer #2 (Remarks to the Author):

The revised manuscript has been greatly improved and it can be published in current state.

Reviewer #3 (Remarks to the Author):

I am satisfied with the science and I am supportive of publication.

However, the scholarly presentation is currently very low. The figures need significant renovation and the text needs major polishing. The title, as one key example, should be revisited so that it holds broader appeal.

Point-by-point response to referees

Reply to Referee 1 and revisions made accordingly:

The manuscript is acceptable with author's revision.

We truly thank the reviewer for the agreement of acceptance with this version!

Reply to Referee 2 and revisions made accordingly:

The revised manuscript has been greatly improved and it can be published in current state.

We are grateful to the reviewer's comments and really appreciate the agreement of publication with this version!

Reply to Referee 3 and revisions made accordingly:

I am satisfied with the science and I am supportive of publication.

However, the scholarly presentation is currently very low. The figures need significant renovation and the text needs major polishing. The title, as one key example, should be revisited so that it holds broader appeal.

We sincerely thank the reviewer for the support of publication and the comments are really helpful to improve the quality of this article. According to the suggestions from the reviewer and editor, we have renovated the figures that the original Fig. 1 is split into two new figures, which is more intelligible for readers. Meanwhile, the English language of the main text has been polished and modified carefully for clarity and readability, which is expected to reach a high level. Furthermore, the title has been changed into “Systematic design of superaerophobic nanotube-array electrode comprised of transition-metal sulfides for overall water splitting”, which is considered to hold broader appeal.